# Altitudinal Variations in Coniferous Vegetation and Soil Carbon Storage in Kalam Temperate Forest, Pakistan

**DOI:** 10.3390/plants14101534

**Published:** 2025-05-20

**Authors:** Bilal Muhammad, Umer Hayat, Lakshmi Gopakumar, Shuangjiang Xiong, Jamshid Ali, Muhammad Tariq Badshah, Saif Ullah, Arif UR Rehman, Qun Yin, Zhongkui Jia

**Affiliations:** 1State Key Laboratory of Efficient Production of Forest Resources, Beijing Forestry University, Beijing 100083, China; yousafzaibilal1992@yahoo.com (B.M.);; 2Engineering Technology Research Center of *Pinus tabuliformis* of National Forestry and Grassland Administration, Beijing Forestry University, Beijing 100083, China; 3State Key Laboratory of Integrated Management of Pest Insect and Rodents, Institute of Zoology, Chinese Academy of Sciences, Beijing 100101, China; oomarcassi6116@gmail.com; 4Post Doctoral Fellow, National Centre for Aquatic Animal Health CUSAT, Ernakulam 682016, Kerala, India; 5Forest Management, School of Forestry, Beijing Forestry University, Beijing 100083, China; 6Key Laboratory of Soil Ecology and Health in Universities of Yunnan Province, Yunnan University, Kunming 650500, China; 7Aerospace Information Research Institute, Chinese Academy of Sciences, University of Chinese Academy of Sciences, Beijing 100083, China

**Keywords:** climate change, forest ecosystem, regeneration, species distribution, vegetation dynamics

## Abstract

Understanding the complex interplay among altitudinal gradients, tree species diversity, structural attributes, and soil carbon (C) is critical for effective coniferous forest management and climate change mitigation. This study addresses a knowledge gap by investigating the effects of altitudinal gradient on coniferous tree diversity, biomass, carbon stock, regeneration, and soil organic carbon storage (SOCs) in the understudied temperate forests of the Hindu-Kush Kalam Valley. Using 120 sample plots 20 × 20 m (400 m^2^) each via a field inventory approach across five altitudinal gradients [E1 (2000–2200 m)–E5 (2801–3000 m)], we comprehensively analyzed tree structure, composition, and SOCs. A total of four coniferous tree species and 2172 individuals were investigated for this study. Our findings reveal that elevation indirectly influences species diversity, SOCs, and forest regeneration. Notably, tree height has a positive relationship with altitudinal gradients, while tree carbon stock exhibits an inverse relationship. Forest disturbance was high in the middle elevation gradients E2–E4, with high deforestation rate at E1 and E2. *Cedrus deodara*, the dominant species, showed the highest deforestation rate at lower elevations (R^2^ = 0.72; *p* < 0.05) and regeneration ability (R^2^ = 0.77; *p* < 0.05), which declined with increasing elevation. Middle elevations had the highest litter carbon stock and SOCs values emphasizing the critical role of elevation gradients in carbon sink and species distribution. The regeneration status and number of trees per ha in Kalam Valley forests showed a significant decline with increasing elevation (*p* < 0.05), with *Cedrus deodara* recording the highest regeneration rate at E1 and *Abies pindrow* the lowest at E5. The PCA revealed that altitudinal gradients factor dominate variability via PCA1, while the Shannon and Simpson Indices drives PCA2, highlighting ecological diversity’s independent role in shaping distinct yet complementary vegetative and ecological perspectives. This study reveals how altitudinal gradients shape forest structure and carbon sequestration, offering critical insights for biodiversity conservation and climate-resilient forest management.

## 1. Introduction

Species diversity is influenced by elevational variation in vegetation diversity. When the species diversity reaches their peak performance at the geographic center, it declines due to unfavorable abiotic conditions as they approach distribution limits [1,2,3]. The changes in species diversity can be attributed to various ecological gradients [4,5]. The direct climatic factors (light, temperature, precipitation, and evaporation) also vary with these factors and habitat types [6,7]. For instance, it is a fact that Beech forests thrive on foggy northern and southern slopes [4,6]. Certain researchers have proposed the hypothesis of uneven abiotic stress limitation, where the lower and upper limits of species distribution in a habitat are influenced by biotic and abiotic factors, respectively [7,8,9]. Previous literature has investigated changes in species diversity and population dynamics along the elevation gradient to understand the changes in these fluctuations, which is also significant for understanding climate-induced distribution changes often assumed without direct examination [10,11,12]. The topographical factors like elevation, aspects, and slope shape can be related to changes in vegetation. The influence of altitudinal gradients on the spatial distribution of species diversity is widely recognized owing to their direct correlation with diverse environmental parameters that afford nuanced ecological circumstances [13,14]. These assumptions, however, were not thoroughly investigated in relation to structural attributes, regeneration potential, and SOCs in the Hindu Kush region, which has been experiencing severe stress in high-altitude regions from climatic variability in light of global climate change.

The growth of flora is an important factor related to its species distribution. Among the various components, the most crucial component in mountainous landscapes which primarily affects habitat, climate, and flora is the altitude, and the interplay of variables such as aspect, soil types, and slope gradients significantly influences ecological dynamics [15,16]. It is predicted that the vegetation would move to higher elevation zones due to elevation-induced rise in the temperature at lower altitudes [17] which is proved in the light of global warming that has caused disturbed altitudinal ecological systems, leading to changes in plant life patterns and a general pole wards migration of the species distribution [18]. Climate plays a fundamental role in controlling vegetation distribution patterns [19]. In the Hindu-Kush Himalayan region, several studies [20,21] have also investigated the variation of tree growth with elevation in terms of diversity, regeneration, biomass, and C stock in, and it was seen that at 1600–1900 m and 3100–3400 m, biomass was at its peak in the region. Another study reported that the highest vegetation biomass and C content were discovered in the northern part of the lower elevation belt in the Hindu-Kush range [22,23,24]. These highly variable data for the above ground biomass (AGB), tree carbon stock, structural attributes, regeneration potential, and SOCs for different forest ecosystem across altitudinal gradients highlight the importance of site-specific measurement of parameters with predictive capacity for a good approximation of forest structural composition and C contents in different ecosystems. The investigation of carbon inventories and biomass along elevational gradients in the Kalam Valley, Khyber Pakhtunkhwa, Pakistan, is crucial, as these valleys provide a unique environment to study the relationship between elevation and changes in biomass and carbon stocks. Species richness, evenness and heterogeneity significantly impact the structure and function of forests and are important markers of sustainable forest management. The two main components of biodiversity, species richness and species diversity, are important factors for forest conservation, management, and regeneration capacity and are intimately linked to ecosystem stability in addition to the structure and function of the forests [24,25,26,27,28].

The tree species diversity and richness are also influenced by anthropogenic factors which are a major component to be considered in any forest management related research. One of the most primitive and still prevalent exploitation of humans on forest resources is deforestation [29] that frequently causes habitat fragmentation and altered ecological processes, such as land-use changes, urbanization, deforestation, and agricultural practices. By destroying habitats, upsetting soils, and dividing forests, logging can have a major negative influence on the diversity of tree species. Even though sustainable logging techniques like selective harvesting can be helpful in maintaining biodiversity logging that is not properly managed may result in long-term ecological changes and the loss of essential ecosystem services. The effects can be direct or indirect which can affect the ecosystems and the related flora/ fauna. The direct effects include soil disturbance that causes compaction, erosion, and fertility loss, as well as the removal of vegetation, which alters forest structure and habitats. These impacts raise the possibility of landslides and soil erosion in mountainous areas. Furthermore, logging frequently results in forest fragmentation, which isolates wildlife populations, and it can worsen water quality by increasing runoff and sedimentation in rivers. By decreasing canopy cover, logging indirectly modifies microclimates, influencing light, humidity, and temperature, which puts stress on surviving plant species. Additionally, it can alter hydrological cycles by altering water flow, disturb ecosystems, and create favorable conditions for invasive species, all of which contribute to a further decline in biodiversity.

With its varied forest types, steep altitudinal gradients, and predominance of coniferous species, the Kalam Valley in Northern Pakistan—represents an area of biological and ecological significance. Understanding how elevation affects soil carbon content and forest growth patterns is essential for sustainable forest management, especially in light of the mounting pressure from anthropogenic activities and climate change. Because of the Kalam Valley’s abundant coniferous cover and the lack of data specific to the area that connects altitudinal variation to ecological and carbon-related processes, this study is especially warranted there. Even though there are many studies on forest species diversity, there is a lack of integrated information on the forest structure, species diversity, species composition, and C storage [30] despite the dry temperate forests’ ecological significance. By examining the complex connections between altitudinal gradients, conifer tree growth patterns, ground C composition and SOCs within the dry temperate forests of Northern Pakistan, this study offers a novel point of approach by providing a piece of integrated information on forest structure, tree species diversity, and C storage. Although the impact of altitudinal gradients on plant distribution is well known, this study focuses exclusively on conifer trees as this region has the dominance of conifers to better understand their response to different elevational conditions. This region has been experiencing severe stress resulting from climatic variability (i.e., increased temperature fluctuations, altered precipitation patterns) at high altitudes, surpassing that of the Himalaya and Karakoram ranges [31,32]. This study contributes to regional conservation strategies and global carbon modeling efforts by examining conifer species across elevation gradients and offering important insights into how forest ecosystems in the Hindu Kush may react to environmental changes. The study area does not have any broadleaved trees. As conifers being the main tree groups facing serious threats of poor management and negligence the detailed study of this ecosystem is expected to bring out interesting results that can contribute to ecosystem management. In addition, the study addresses the lack of comprehensive research on the effects of altitudinal gradients on vegetation diversity, biomass, regeneration, and soil organic carbon (SOC) storage in temperate mountainous forest ecosystems, particularly in the Hindu-Kush Kalam valley. There is limited understanding on the influence of elevation on tree species composition, carbon dynamics, and forest structural attributes in this region, which is critical for effective forest management and climate change mitigation. To address these research gaps, this study investigates the influence of altitudinal gradients on tree species diversity, biomass, carbon stock, forest regeneration, and soil organic carbon (SOC) storage in the temperate forests of the Hindu-Kush Kalam valley. The study aimed to evaluate the influence of elevational changes on ground C contents in conifer stands, assess the disturbances, regeneration status, species diversity and distribution of conifer species along different altitudinal gradients; and investigate the role of Above Ground Biomass (ABG) and elevation as drivers of soil carbon storage (SOC) in the dry temperate forests of the Kalam valley. We also tried to understand the influence of anthropogenic and environmental factors on the tree species diversity and distribution in the study area. We hypothesize that increasing elevation is associated with a decline in the growth rate and carbon stock of conifer tree species due to environmental constraints such as lower temperatures, reduced soil fertility, and shorter growing seasons. Consequently, higher elevations are expected to support reduced tree development, biodiversity, and soil carbon storage compared to lower elevations. We also hypothesized that the diversity and spatial distribution of trees in a mountainous forest ecosystem are influenced by environmental and anthropogenic factors like logging.

## 2. Results

### 2.1. Growing Stock and Biomass

The tree species at various elevation gradients in our study was represented by *Cedrus deodara*, *Pinus wallichiana*, *Picea smithiana*, and *Abies pindrow*. As the study area lies within the dry temperate zone, so these are the main tree species occupying the area’s forest cover.

A multiple linear regression model was used to examine the influence of site, species, elevation, slope, aspect, and structural variables (basal area and volume) on tree carbon stock (TCS) [Table 1]. The model was statistically significant overall (*p* < 0.001), with several predictors contributing significantly to the variation in TCS. Among the structural variables, tree volume (B = 2.060, *p* < 0.001) and basal area (B = 59.148, *p* < 0.001) had the strongest positive effects on TCS. In terms of species effects, Fir, Spruce, and Blue Pine all showed significant negative impacts on carbon stock compared to the reference species (Diar), with Fir having the largest negative coefficient (B = −65.847, *p* < 0.001). Site-related factors also influenced TCS. The location Usho had significantly lower TCS compared to Utror (B = −9.456, *p* < 0.001). Similarly, elevation levels E2, E3, and E5 were associated with significantly lower TCS compared to the base elevation E1 (*p* < 0.05). Both Aspect_G2 (S/SE/SW; B = −3.387, *p* = 0.01) and_G3 (E/W; B = −4.484, *p* = 0.005) showed significantly lower TCS compared to the G1 (N/NE/NW). Slope categories, including gentle, moderate, and steep, did not show significant effects on TCS (*p* > 0.05). Multicollinearity diagnostics showed that VIF values were mostly within acceptable limits, though basal area and volume showed higher VIFs (10.946 and 10.656, respectively), indicating some degree of collinearity. All structural attributes of each species and tree C variability along an altitudinal gradient are detailed in Appendix A.

### 2.2. Association Between DBH, Tree Height, and Stand Density at Different Elevations

The highest DBH value (53.34 ± 2.69 cm) was recorded for *Picea smithiana* at E5 and the lowest (27.88 ± 1.04 cm) for *Cedrus deodara* at E1. The maximum height (23.33 ± 0.51 m) was seen for *Picea smithiana* at E5 and the minimum (11.14 ± 0.31 m) for *Pinus wallichiaana* at E1 [Appendix A]. Tree basal area and volume showed significant reduction with the upward shift in elevation. The DBH and tree height showed a positive relationship at various elevations (*p* < 0.05). The highest correlation values between DBH and tree height were observed at E2 and E5 (R^2^ = 0.83, *p* < 0.05) [Figure 1]. Tree DBH and height were positively influenced by elevation while tree biomass and carbon stock were negatively influenced by elevation.

The DBH and tree height showed positive relationship at various elevations (*p* < 0.05). A negative correlation between DBH and stand density was also observed (R^2^ = −0.93; *p* < 0.05). The highest correlation values between DBH and stand density were observed at E3 and E4 (R^2^ = −0.93, *p* < 0.05). A significant correlation existed between DBH, tree height, and stand density, particularly at higher elevations. Tree volume (m^3^/ha) showed a significant positive relationship with basal area (m^2^/ha) (*p* < 0.05), with the strongest correlation at E3 (R^2^ = 0.94; *p* < 0.05) and significant correlations at E2 (R^2^ = 0.94; *p* < 0.05), E5 (R^2^ = 0.93; *p* < 0.05), E1 (R^2^ = 0.86; *p* < 0.05), and E4 (R^2^ = 0.93; *p* < 0.05) [Appendix A].

### 2.3. Association of Species Dominance and Species Diversity with Elevation Gradients

The Association of the Relative Density (RD), Relative Abundance (RA), Relative Frequency (RF) and Important Value Index (IVI) of different tree species with elevation gradient was significant (df = 76, F = 4.78, *p* = 0.0001; *p* < 0.05). At all the elevation gradients, the dominant species was *Cedrus deodara* and the co-dominant species was *Abies pindrow* with either *Cedrus deodara* or *Pinus wallichiana* as the co-dominantspecies [Figure 2]. All species were significantly negatively influenced by elevation gradients for tree, relative density, relative frequency, relative dominance and IVI (*p* < 0.05) [Figure 2]. *Cedrus deodara* and *Pinus wallichiana* were the dominant species most negatively influenced by elevation (R^2^ = 0.81 and 0.82; *p* < 0.05). However, *Picea smithiana* was the least negatively influenced (R^2^ = 0.57; *p* < 0.05).

The Shannon index value ranged from 1.18 ± 0.02 at E1 to 1.09 ± 0.01 at E5 [Table 2]. The Simpson Index ranged from 3.28 ± 0.04 at E1 to 2.94 ± 0.02 at E5. Species evenness varied from 0.91 ± 0.01 at E1 to 0.87 ± 0.01 at E5. ANOVA showed a significant effect of elevation on species diversity (*p* < 0.05), the Tukey’s post hoc test showed a significant variance among means, and the variability in the Shannon, Simpson index, and species evenness explained by elevation was significant to address their association. The Shannon and Simpson index significantly indicated that the diversity reduced with increasing elevation [R^2^ = 0.72, 0.73, *p* < 0.05]. Whereas Species evenness was positively correlated with elevation [R^2^ = 0.87; *p* < 0.05]. However, no significant association was observed between elevation gradients and Species Richness. The Shannon Index, Simpson index, Species Evenness and species richness showed varying trends to elevational gradients.

### 2.4. Association of Litter, Deadwood, and SOCs with Elevation Gradients

The litter carbon stock displayed significant variation (*p* < 0.05), with mean values ranging from 1.76 ± 0.2 to 2.22 t C/ha across the elevations [Figure 3a]. Specifically, elevation E1 and E3 emerged as the central point of interest, with the highest average litter carbon stock of 2.22 t C/ha. Furthermore, there was a considerable connection between elevation and litter carbon stock distribution (R^2^ = −0.71) [Figure 3a]. These findings highlight the significant influence of E1 and E3 in shaping the distribution of litter carbon stock. The investigation into deadwood carbon stock revealed notable findings indicating substantial differences (*p* < 0.05) across different elevations (20.31 ± 0.43 t C/ha at E1, 20.4 ± 0.41 t C/ha at E2, 20.3 ± 0.41 t C/ha at E3, 19.97 ± 0.38 t C/ha at E4, and 19.81 ± 0.37 t C/ha at E5) respectively. Elevation gradient E2 showed a distinct characteristic by displaying the highest deadwood carbon stock of 20.4 ± 0.41 t C/ha. The observed data change was statistically significant (df = 4, F = 11.39, *p* = 0.04; *p* < 0.05). There was negative correlation observed between deadwood carbon stock concentration and elevation gradients (R^2^ = −0.71, *p* < 0.05) [Figure 3a]. These findings underscore the important influence of E2 on the deadwood carbon stock. The SOCs at depths 10 cm, 20 cm, and 30 cm across different altitudes revealed a significant difference (df = 12, F = 8.77, *p* = 0.001; *p* < 0.05). The SOCs at Elevation E3 consistently displayed the highest values throughout different soil depths. A distinct pattern was observed, with the highest soil carbon stock at 10 cm depth, reaching 87.84 ± 8.79 tC/ha. Although the lowest total SOCs value was calculated at E5 (108.77 ± 12.34 tC/ha) but there was no statistically significant correlation found (R^2^= −0.32, *p*> 0.05). The observed fluctuation in elevation demonstrated statistical significance (*p* < 0.05), as evidenced by strong correlations with R^2^ values (0.84, 0.97, and 0.96 at depths 10 cm, 20 cm, and 30 cm, respectively). These findings confirm the significant influence of distinct elevations on the spatial distribution of LCS, DWCS, and SOCs within forest ecosystems.

### 2.5. The Distribution of C Pools over Various Elevations

Significant variations were observed in the C stocks of individual ecosystem factors at different elevations gradients E1 to E5 (df = 24, F = 2.88, *p* = 0.04; *p* < 0.05) [Figure 4]. The levels of above-ground carbon stock (AGCs) varied significantly between altitudes (250.58 ± 21.43 at E1 to 190.27 ± 18.43 t C/ha at E5), indicating that the AGCs contained a distinct distribution of C across altitudinal gradients. Similarly, below-ground carbon stock (BGCs) at different elevations revealed significant differences (44.89 ± 7.32 at E1 to 39.81 t C/ha at E5), highlighting the critical role that elevation plays in determining BGCs. The variations in sapling C stock (SSCs) were also substantial (95.27 at E1 to 88.21 t C/ha at E5), suggesting that the growth and establishment of juvenile vegetation is influenced by altitude. The variation in litter C stock (LCs) from 2.22 ± 0.11 at E1 and E3 to 1.70 t C/ha at E5 across different elevations indicated distinct organic matter decomposition rates. The variation in dead wood C stock (DwCs) values 20.40 ± 1.46 to 19.81 ± 0.96 t C/ha at E5 underscored the impact of altitude on the decomposition and accumulation of woody materials. In addition, notable variations were observed in the SOCs with the values ranging from 221.06 ± 24.58 at E3 to 108.77 ± 19.81 t C/ha at E5, underscoring the influence of altitude on soil C dynamics. The total C stock (TCs) was calculated highest at E3 (622.58 ± 38.43 t C/ha) and the lowest at E5 (477.28 ± 27.78 tC/ha) that demonstrated the overall effect of altitude on C storage [Figure 4]. The findings of two-factor ANOVA, followed by Tukey’s post-hoc test, emphasized the importance of elevation in influencing C stocks, as each component exhibited a distinct response to the elevation gradient.

### 2.6. Regeneration Ability, Number of Trees per Ha and Deforestattion Status of the Conifer Forestat Various Elevation Gradients

The regeneration ability of the forests significantly varied at different elevations in the study area (df = 57, F = 4.91, *p* = 0.00001; *p* < 0.05) and was estimated from the number of seedlings, saplings and trees [Figure 5]. In the seedling layer of vegetation, the average number varied from 18 ± 1.01 to 181 ± 11.21 seedlings/ha in low disturbance zone and 13 ± 0.89 to 127 ± 6.56 seedlings/ha in high disturbance zone with lowest values at E5 and highest values at E1 (*p* < 0.05). The sapling density varied from 7 ± 0.31 to 91 ± 5.32 saplings/ha in low disturbance zone and 3 ± 0.09 to 64 ± 2.34 saplings/ha in high disturbance zone with lowest values at E3 and highest values at E1 (*p* < 0.05). The tree density varied from 22 ± 2.13 trees/ha to 335 ± 13.43 trees/ha in low disturbance zone and 15 ± 3.01 trees/ha to 90 ± 8.91 trees/ha in the high disturbance zone with lowest values at E5 and highest values at E1 (*p* < 0.05). In the low disturbance zone the tree species such as *Abies pindrow* showed relatively lower regeneration while *Cedrus deodara* showed relatively high regeneration, on the other hand in high disturbance zone *C. deodara* showed lower generation while *A. pindrow* showed higher regeneration (*p* < 0.05).

The effect of elevation gradients on regeneration status and number of trees per ha showed a significant negative correlation (*p* < 0.05). A negative relation between elevation gradients and seedlings was seen with the highest relationship for *Picea smithiana* (R^2^ = 0.83; *p* < 0.05) and the least value for *Pinus wallichiana* (R^2^ = 0.75; *p* < 0.05); similarly, a negative relation between elevation gradients and saplings was seen with the highest relationship for *Cedrus deodara* (R^2^ = 0.80; *p* < 0.05) and the lowest for *Pinus wallichiana* (R^2^ = 0.68; *p* < 0.05); the relation between elevation gradients and number of trees per ha was calculated highest for *Cedrus deodara* (R^2^ = 0.81; *p* < 0.05) [Figure 6a]. The elevation gradient and deforestation rate were observed to be correlated negatively (*p* < 0.05), with the highest number of cut trees 34 ± 4.56 trees/ha of *Cedrus deodara* at E1 and the lowest deforestation rate 20 ± 2.78 trees/ha at E5. Highest association value (R^2^ = 0.76; *p* < 0.05) was calculated for the *Pinus wallichiana* and the lowest (R^2^ = 0.72; *p* < 0.05) was calculated for *Cedrus deodara* respectively [Figure 6b].

### 2.7. PCA of Environmental and Vegetative Variables

The PCA investigation uncovers relationship among ecological variables, regeneration and biodiversity parameters [Figure 7]. The negative correlation identified in elevation (red) corresponds to ecological variables such as Tree Carbon Stock, Deadwood Carbon, Soil Organic Carbon (SOC), Regeneration, Shannon Index, Simpson Index and Species Richness. While there is a positive correlation between elevation and the tree variables (Tree Height, DBH, Tree Volume, and Basal Area).

## 3. Discussion

Our study found a positive relationship between tree C stock and variables such as elevation, aspect, DBH, basal area, tree height, and volume. These findings are consistent with previous research, which highlights that tree growth is significantly influenced by rainfall, altitude, and plant species composition [33,34]. Trees at higher elevations generally receive more sunlight, though tree height typically decreases with elevation, reflecting variations in nutrient availability, climate, and plant competition [35,36,37]. In our study, well-drained soil and higher organic C levels at lower elevations suggest sufficient nutrient supply for tree growth. Although reduced growth at higher altitudes was expected due to lower temperatures and limited nutrient availability, this trend was not observed in our study.

The tree growth at various elevations of the study area could be supported by local ecological conditions like deadwood accumulation and increased litter or changes in species composition favoring species that are resource-efficient or stress tolerant. Organic C may also play a role in nutrient binding and plant growth. Furthermore, the decrease in C stock with elevation could be attributed to less favorable soil or atmospheric conditions at higher altitudes [38]. This contrasts with findings from a previous study [39,40], which reported an increase in biomass and C stock with elevation.

The above findings emphasize the need to take account of local ecological drivers and in addition to broad climatic gradients and complexity of forest growth responses to elevation. Microclimate, soil properties, and species-specific functional traits can all have a significant impact on forest productivity in diverse mountainous environments. In contrast to a number of earlier studies that have documented a general decline in tree growth and carbon stock at higher elevations, our study found a positive relationship between elevation and tree growth variables (such as tree height, DBH, and carbon stock). While most studies indicate that trees are generally under stress at higher elevations due to harsher climatic conditions, decreased nutrient availability, and cooler temperatures, which results in stunted growth and lower biomass, our findings imply that these anticipated limitations might be reduced by elements like more sunlight exposure, nutrient cycling through the accumulation of deadwood and litter, and species composition changes. Furthermore, contrary to our findings that tree height rose with altitude, other studies have found that tree biomass and height decrease with elevation, especially in temperate and tropical regions This divergence may be attributed to site-specific factors such as soil fertility, microclimatic variations, and the species composition of the study area, which can either enhance or limit tree growth depending on local conditions. Additionally, although some studies have documented increases in biomass and carbon stock at higher elevations as a result of improved nutrient availability or particular plant adaptations, the overall pattern is still very context-dependent, with differing findings based on study methodology, elevation range, and geographic region. Furthermore, the increase in DBH values with altitude for trees with a DBH less than 20 cm is likely due to more sunlight availability in deforested areas than the forested areas where trees face less competition [Figure 1]. Similar findings have been observed globally, where high canopy openness due to deforestation enhances light penetration [41,42]. Although competition for growth influences lower altitudes, competition for light is minimal at higher altitudes and in less dense forests, permitting the trees to grow taller [43,44]. This relationship was also evident in our study. At higher alpine elevations, cooler temperatures and higher soil moisture may contribute to reduced growth, with a reduction of height at alpine elevations possibly resulting from lower plant metabolism [45,46,47]. However, the negative effects of elevation on stand density are more pronounced in single species stands than multispecies stands. In this multispecies study, the plants did not show a pronounced height reduction with elevation, which contrasts with the general observation that trees grown in the open conditions tend to be shorter for a given DBH compared to those grown in closed conditions. Our findings are supported by other research which shows that tree growth does not follow a universal growth pattern and differs based on environmental factors and microclimate [48,49]. However the results align with our study objectives highlighting the need for protecting the existing biodiversity that influences the stand density.

Different tree species exhibited similar relationships with elevation. The reduced numbers of *Cedrus deodara*, *Picea smithiana*, *Abies pindrow*, and *Pinus wallichiana* with increasing elevation and the negative relationship in the number of seedlings, saplings, and trees for all species with elevation is likely due to more favorable conditions at lower altitudes. Seedling density generally increases with lower altitude, with higher survival rates in gap areas due to better germination conditions [50,51]. This trend supports our hypothesis regarding the influence of elevational gradients on climate change mitigation.

Species-specific sampling revealed *Cedrus deodara* as the dominant species at all elevations, with *Abies pindrow* or *Pinus wallichiana* as codominants [Figure 2]. The significant negative relationship with elevation may be due to reduced soil nutrient and water availability at higher altitudes. Similar variations in seedling, sapling, and tree density across altitudinal gradients have been reported in Nepal and Southern Ethiopia [47,50,51,52,53,54]. The negative association between the Shannon-Weiner index and species richness with altitude [Table 2] may result from reduced habitat diversity at higher elevations due to unfavorable climatic and soil conditions. Conversely, the positive relationship of the Simpson index with altitude suggests increased community diversity at higher elevations. This indicates that plant communities adapted to high-altitude conditions become more prevalent with increasing height, as supported by other studies [53,54]. Our results reinforce the hypothesis that elevation significantly influences the floral diversity of tree species in mountainous regions.

Variations in litter, deadwood, and SOCs showed distinct patterns, with the highest values at mid-altitudes. The dependency of deadwood and forest litter on site features, altitudinal gradients, forest type, and forest management supports our findings [55,56,57]. Similar observations have been reported in the forests of the Hindu-Kush Himalayan region [58,59]. Tree layer biomass is generally higher in coniferous forests. Deadwood is a crucial component of the forest C pool, with its decay influenced by wood type, size, properties, and regional climatic and soil conditions [60,61,62]. The highest values for organic C, litter, and SOCs at mid-altitudes can be attributed to higher erosion rates and less favorable soil and atmospheric conditions at higher altitudes. Slope plays a significant role in this process, as litter washes down during the rainy season and deposits at less steep slopes, typically at mid-altitudes. This also affects C storage at different elevations. Dominant species like *Cedrus deodara* have shown an increased ability for C storage, as corroborated by several studies in the Hindu-Kush range [63,64].

As altitude increased, a decrease in the above-ground carbon (AGC) stock was noted, most likely as a result of changes in slope, soil erosion rates, and related site conditions. This pattern is consistent with results from other studies that show modest increases at mid-elevation ranges and lower AGC on steeper slopes [63,64,65,66]. Reduced AGC may also result from logging activities, which were visible in the study area. This would emphasize how anthropogenic disturbance shapes carbon dynamics. Apart from these human and physical factors, low temperatures, decreased transpiration, and poor soil quality frequently limit plant growth at higher elevations, and temperature fluctuations can raise respiration costs [67,68,69]. The importance of altitude in determining patterns of carbon storage is further highlighted by variations in below-ground carbon (BGC) across the elevation gradient. Significant variations in BGC imply that elevation is a major factor in determining the contributions of soil organic matter and root biomass. Similar to this, variations in the carbon stock of saplings along the gradient suggest that elevation influences the establishment and growth of juvenile vegetation, most likely by influencing the soil and microclimate. Elevation also affected the litter carbon stock, which reflected variations in the input of organic matter and the rates of decomposition. Similarly, variations in the carbon stock of deadwood across elevations show how altitude affects the buildup and decomposition of woody materials. The impact of altitude on below-ground carbon processes is further demonstrated by variations in soil organic carbon (SOC), which are probably influenced by temperature, moisture, and microbial activity gradients. All things considered, the observed variation in the overall ecosystem carbon stock across elevation bands emphasizes the interplay between site conditions, species composition, and altitude. The hypothesis that elevation has a significant impact on all components of the carbon stock, each of which responds differently to environmental gradients, is further supported by the results of the single factor

Seedling and sapling density densities differed considerably in both low and high disturbed zones across different elevations. These changes in density may be related to variables like altitude and anthropogenic disturbances [70]. We observed significantly lower seedling and sapling density in high disturbance areas compared to low disturbance areas which indicated that anthropogenic disturbances, including tourism, deforestation, and grazing in the study region, likely contribute to the overall low density of seedlings and saplings [71].

The high regeneration status observed in low disturbance zone reveals the positive influence on low/no disturbance on the maintenance of forests. This can be also linked to elevation, as a significant negative relationship between regeneration status and elevation was observed, as evidenced by the lower number of seedlings at higher elevations [Figure 5 and Figure 6]. This may be due to the lack of adequate sunlight received by seedlings in the presence of tall trees, as well as reduced nutrient and water availability [72]. Higher soil nutrient levels and moisture at lower elevations explain the higher number of seedlings at E1. The reduced logging of trees at higher elevations might be due to challenging topographical features like slopes, which impede access to loggers. Our findings are consistent with previous research [50,51], which indicated higher basal area and species diversity in distant forest areas compared to those near forest edges. In short, the results point out the need for proper management of the edge areas so as to ensure biodiversity loss prevention and environmental degradation like landslides and soil erosion.

*Cedrus deodara*’s remarkable regeneration status at all elevations may be related to its ecological resilience and significance in mitigating the effects of climate change. C. deodara is a species that is well-known for its high biomass productivity and substantial contribution to soil carbon sequestration [50,52]. As such, it is essential for improving ecosystem carbon storage. Our results highlight how crucial dominant species are for promoting natural regeneration and sustaining carbon dynamics via secondary succession. Nevertheless, continuous deforestation, especially at lower elevations, poses a major threat to the sustainability of forests despite their capacity for regeneration. One of the main causes of Pakistan’s natural forest decline is still deforestation [36,72], which breaks up vegetation continuity and reduces biodiversity in these areas. Furthermore, lower elevation forest degradation can worsen soil erosion, decrease soil fauna populations, and lead to a decrease in soil organic carbon, all of which can lower soil fertility and quality, which are essential for sustaining a variety of productive vegetation [69,71,72]. These ecosystems may become less diverse and carbon-poor as a result of this degradation, which would lessen their ability to mitigate climate change. The necessity of giving lower elevation vegetation conservation top priority is further highlighted by the lower regeneration rates seen at higher elevations. This promotes the growth of self-sustaining forest ecosystems by preserving biodiversity in the lowlands and enhancing the resilience and regeneration of higher elevation forests.

The positive correlation between elevation and tree variables as revealed by PCA means that structural features of trees tend to be more noticeable at higher elevations. The adaptive traits of species that flourish at higher elevations, including bigger trees that suppress the growth of young seedlings could be the reason for this. The Regeneration Index and the Shannon Index, along with Species Richness, suggest that species establishment and biodiversity may not be directly correlated with elevation, but rather influenced by other environmental factors, indicating less links. Highlighting that elevation predominantly influences ecological structure, the presence of central axis lines offers a definitive reference and underscores their significance as a primary ecological element in this study area. These findings shed light on how elevation affects forest composition and ecosystem services by highlighting the complex relationship between forest dynamics and environmental gradients

## 4. Materials and Methods

### 4.1. The Study Area

The study was conducted in Kalam Valley (KV), District Swat, northern Pakistan, located approximately 160 km north of Islamabad. Swat is a mountainous green valley lying between 34–40° to 35′ N and 72–74° to 60′ E, with an elevation between 2000 to 3900 m above sea level. The total area is approximately 1600 km^2^ [Figure 8].

The annual climatic pattern shows fluctuating atmospheric conditions. The climate is very cold during winter (November, December, January, and February) with frequent snowfall and moderate during summer (June, July, and August). The annual climatic pattern shows fluctuating atmospheric conditions. The region has frequent rainfall in March–April, with relatively dry summer (June–August) and autumn (September and October) seasons. The temperature of the study area varies from 7.48 °C to 30.94 °C. The precipitation ranges from 30.23 mm to 272.09 mm and the relative humidity varies from 31.35% to 69.38%. The lowest reported temperature is during January (−4 °C) and highest in July (38 °C). The lowest precipitation and humidity is during December and the highest during August [Table 3]. The soil is sandy loam type with good drainage.

The vegetation of the area represents a mixture of evergreen vegetation types. About 160 km^2^ of the forest area is dominated by silver fir (*Abies pindrow*), Spruce (*Picea smithiana*), Deodar (*Cedrus deodara*) and Kail or Blue pine (*Pinus wallichiana*). Due to deforestation, the forests in this hilly area are not thick and have many gaps.

### 4.2. Data Collection

#### 4.2.1. Field Inventory and Sampling

A field inventory was conducted from 5 July to 15 August 2022 for the collection of field data. We selected a representative mountain from all the surrounding forested mountains having most of the major ecological, edaphic, and topographical features in common. The dry temperate forest zone (2100–3000 m) was divided into five altitudinal zones with coding from E1 to E5 (E1 = 2000–2200 m; E2 = 2201–2400 m; E3 = 2401–2600 m; E4 = 2601–2800 m; E5 = 2801–3000 m) along a gradient with an interval of 200 m. We established 120 sample plots (79 in Utrur valley and 41 in Usho forest) of 20 × 20 m (400 m^2^) each, and a total of 16,000 m^2^ area was sampled at each altitudinal zone to analyze the tree structure and composition [Appendix A]. For data collection on trees, saplings, and seedlings, the diameter at breast height (DBH) of trees >15 cm was measured in a 20 × 20 m square plot. The number of stems for saplings DBH < 15 cm was measured in the square plot of 6 × 6 m, and for seedlings, the data collection was from a 2.5 × 2.5 m square plot [Figure 9].

Disturbance levels were classified based on anthropogenic pressure indicators such as proximity to human settlements, intensity of grazing, and extent of deforestation. Forest areas between 2000–2400 m was categorized as high disturbance zones due to their close proximity to communities and nomadic territories, characterized by frequent livestock grazing and significant deforestation. In contrast, areas above 2400 m were defined as low disturbance zones, as they were more remote, experienced minimal human presence, low grazing pressure, and limited deforestation.

At each altitudinal zone, 24 sampled quadrats were taken randomly to determine the plant biodiversity. The position of each plot was marked using the Geographical Positioning System (GPS). Tree diameter, height, and stem density were measured in each plot. Tree diameter was measured by calibrated diameter tape at the breast height (DBH) of trees taken at 4.5 feet above the ground from the uphill side. Stand density (number of trees per square meter) in each plot was calculated by counting the number of trees. The vertical distance between the highest tip of the tree and the base of the tree was taken for the tree height and was measured by altimeter (Haga, Gothenburg, Sweden, MPN 43890).

#### 4.2.2. Litter Sampling

Fresh weights were taken from the 1 × 1 m subplots’ litter and leaves. A 100 g of evenly mixed sub-samples were put in a polythene bag and transported to the lab to assess the moisture content, thus allowing calculation of total dry mass and organic matter [73].

#### 4.2.3. Deadwood Sampling

Deadwood biomass was calculated by measuring standing dead wood with branches using a method comparable to the allometric equation for estimating above ground biomass; however, if the dead wood measured <1.3 m in height, its diameter and height were measured as near the top as possible and classified as a logged tree [70,73]. When standing dead wood was leafless, there was a 2–3% reduction for hardwood/broadleaved species and 5–6% for soft-wood/conifer species from their branches [70].

#### 4.2.4. Sampling for Species Diversity

The soil samples were systematically taken at 0–10 cm, 10–20 cm, and 20–30 cm depths in triplicates from each sample plot between 2000 to 3000 m elevations in the forest using a soil auger. The soil at 2100 m elevation had low calcareous content, which increased with altitude (2100–3000 m), leading to pH levels between 5.01 and 7.50.

### 4.3. Calculation of Growing Stock and Biomass

The volume of each tree was estimated from the tree height, basal area, and form factor. The Above ground biomass (AGB Mg/ha) of *Cedrus deodara*, *Pinus wallichiana*, *Picea smithiana*, and *Abies pindrow*, was calculated using the allometric equations (Table 4).

Where *AGB* = Above ground biomass (kg), *D* = Diameter at breast height (cm), *H* = Height (m). Whereas Equation (1) was used to calculate sapling above-ground biomass [73].(1)log⁡AGSB=a+b log⁡(D)
where, *AGSB* = above-ground sapling biomass (kg), *D* = diameter at breast height (cm), *a* = intercept of allometric relationship for saplings [dimensionless], and *b* = slope allometric relationship for saplings [dimensionless].

After calculating the above-ground biomass of the tree, below-ground biomass was calculated using the root-to-shoot ratio (R) conversion method by multiplying aboveground biomass with 0.2 [73]. Total tree biomass was calculated by summing above and below-ground biomass [37]. For tree carbon stock calculations, total tree biomass was multiplied with a conversion factor of 0.47 [73,74,75,76,77].

To calculate the litter biomass, we used Equation (2).(2)$$LB=\frac{W_{field}}{A}\times \frac{W_{sub-sample(dry)}}{W_{sub-sample(wet)}}\times \frac{1}{10,000}$$(3)$$C_{L}=LB\times C_{\%}$$
where, *LB* = Litter biomass t/ha; *W_field_* = weight of fresh field sample of litter sampled within an area of size (g); *A* = size of the area in which litter was collected (ha); *W_sub-sample_*_(*dry*)_ = weight of the oven-dry sub-sample, *W_sub-sample_*_(*fresh*)_ = weight of the fresh sub-sample of litter taken to the laboratory to determine moisture content (g), *C_L_* = total carbon stocks in the dead litter t/ha, *C*_%_ = carbon fraction determined in the laboratory [70,73].

Deadwood carbon stock estimates were derived by deducting two to three percent from each tree’s total above-ground biomass. The biomass value in standing dead wood was calculated using the allometric Equation (4) established in the REDD method [33,38,70,76].

Equation (5) was used to compute total deadwood biomass; to compute deadwood carbon stock, total deadwood biomass was multiplied by a conversion factor of 0.47.(4)SDWD=Σi=0n 13×[D200]2×h×s(5)$$DWB_{T}=SDWB_{Branched}+SDWB_{non-branched}$$
where, *SDWB* = standing deadwood biomass (kg), *h* = length (m), *D* = tree diameter (cm), *s* = specific gravity (g/cm^3^) of wood [value of *s* was used as 0.5 g/cm^3^ as suggested by Hairiah et al. [78], *DWB_T_* = Total Deadwood biomass, *SDWB_Branched_* = standing deadwood biomass of branched wood, and *SDWB_non-branched_* = standing deadwood biomass of nonbranched wood.

### 4.4. Soil Measurement

To determine soil bulk density (BD) the soil was passed through a 2-mm sieve and air dried. The dry weight of the soil divided by the volume of the soil gave the bulk density value (Equation (6)). The soil pH was assessed by mixing it with deionized water at a ratio of 1:2.5 (*w*/*v*). The collected soil samples from an organic layer of soil (0–30 cm) were sent to the laboratory for SOC analysis. SOC was determined using a merged Vario TOC analyzer (Vario TOC, Langenselbold, Germany) by dry combustion at 980 °C in solid mode (De Chem-Tech. GmbH, CleverChem380, Hamburg, Germany). SOCs were calculated using Equation (7).(6)BD=WdVs
where *B_D_* = bulk density, *W_d_* = oven dry weight of the sample (g), and *V_s_* = volume of soil core (cm^3^).SOCs = SOC × BD × SD(7)
where SOCs is the SOC storage (Mg/ha), SOC is the soil organic C content (g/kg), SBD is the soil bulk density (g/cm^3^), and SD is the soil layer thickness (cm). The soil bulk density was determined by Gravimetric method [79,80].

### 4.5. Tree Species Composition, Dominance and Diversity

To calculate the forest floral biodiversity, species diversity was determined through prevalent practices to classify plant communities through indices like the Shannon-Wiener diversity index [81], Pielou’s evenness index [82], Simpson’s diversity index [83], and species richness.

Shannon index (*H*′)(8)$$H^{\prime }=-\sum_{n=1}^{s}\;p_{i}lnp_{i}$$
where *p_i_* = Proportion of individuals of species *i*, *s* = Total number of species.

Pielou’s evenness index (*J*′)(9)$$\begin{array}{c} J^{\prime }=\frac{H^{\prime }}{H_{max}^{\prime }}\\ H_{max}^{\prime }=\ln\left(s\right) \end{array}$$
where *H*′ = Shannon index, *s* = Number of species.

Simpson’s diversity index (D)(10)$$\mathrm{Simpson}\;\mathrm{Index}\;\left(D\right)=1-\sum {pi}^{2}$$
where N = total number of individuals, n = Number of individuals of each species.

Furthermore, in our study, we used IVI data to specify the dominant species as proposed by several studies. The following equations were used for the calculations [31,84];(11)$$\mathrm{Relative}\;\mathrm{density}\;(\mathrm{RDe})=\frac{Number\;of\;individuals\;of\;a\;species}{Total\;number\;of\;individuals}\times 100$$
(12)$$\mathrm{Relative}\;\mathrm{dominance}\;(\mathrm{RDo})=\frac{Total\;basal\;area\;for\;a\;species}{total\;basal\;area\;of\;all\;species}\times 100$$
(13)$$\mathrm{Relative}\;\mathrm{frequency}\;(\mathrm{RF})=\frac{Frequency\;of\;a\;species}{Sum\;of\;all\;frequencies}\times 100$$
Importance Value Index (IVI) = RDo + RDe + RF(14)
where *Frequency of a species* is defined as the number of sampling units in which a particular species occurs, and *Sum of all frequencies* is defined as the total number of sampling units in which all species combined occur (i.e., the *sum of frequencies* for each species).

### 4.6. Total Carbon Stocks 

The C stocks in trees, deadwood, and herbs were assessed by a conversion factor of 0.5 and the total C stock of the whole forest area was calculated using the following equations [84,85,86];(15)$$CS\left(t\frac{C}{ha}\right)=Biomass\times 0.5$$(16)$$Total\;C\;Stock\left(t\frac{C}{ha}\right)=CST+SSCS+SOCs+CSDW+CSL+CSH$$
where CS = C stock, CST = C stock of trees, CSSOCs = Soil organic carbon stock, CSDW = C stock of dead wood, CSL = C stock of litter, CSH = C stock of herbs.

The investigation on the association of litter, deadwood, and SOCs with elevation gradients involved an extensive analysis using a single-factor ANOVA followed by a Tukey’s post hoc test across five elevations: E1, E2, E3, E4, and E5

### 4.7. Regeneration Ability

The level of disturbance of the forests at various elevations (arising from natural and anthropogenic factors) was identified and the elevations were classified as Low Disturbance and High Disturbance areas. The regeneration status of the forest sites under study was estimated based on the level of disturbance. The regeneration status was estimated by measuring the population size of seedlings, saplings, trees and estimating the regeneration status of the tree species from the calculated values at different elevations in least disturbed and highly disturbed areas [86]. Further ANOVA and Tukey post-Hoc test were used to check the data with respect to elevation for each species and to check the significance difference within and between the groups.

### 4.8. Statistical Analysis

The data homogeneity (Levene’s test) and normality (Shapiro–Wilk test) of the variables were tested prior to statistical analysis. The results were calculated using data from 120 sampling plots by assessing six response variables, namely the number of tree species, stand density, tree diameter, tree height, tree basal area, and tree biomass. To assess the effects of site, species, topography, and stand structure on tree carbon stock (TCS), multiple linear regression (MLR) analysis was performed. All categorical variables were dummy coded using reference coding. For each categorical variable with k levels, (k − 1) dummy variables were created, with one level serving as the reference category. Specifically, Location had two categories, with Utror as the reference and Usho as the dummy variable. Elevation (E1–E5) was coded using E1 as the reference, and dummies were created for E2 through E5. Slope categories included Plane (reference), Gentle, Moderate, and Steep. The aspect was grouped into three directional clusters: Group1 (North, NE, NW; reference), Group2 (South, SE, SW), and Group3 (East, West). Species included *Cedrus deodara* (reference), *Pinus wallichiana*, *Picea smithiana*, and *Abies pindrow*. In addition to these categorical predictors, basal area and volume were included as continuous independent variables. The reference categories were selected based on ecological relevance and representation. The final model included all dummy variables along with the continuous predictors to estimate their effect on TCS. Linear regression models were used to evaluate the regeneration status, deforestation status, importance value index (IVI), and species diversity indices of different coniferous tree species along with altitudinal gradients. One-way analysis of variance (ANOVA), and 2-way ANOVA along with all pair-wise comparisons and Tukey’s post-Hock tests in various parameters, was used to test the significant differences of the mean values at different elevation gradients. Principal Component Analysis (PCA) was performed to reduce dimensionality and identify the key drivers of variability in the dataset, with elevation factor, Shannon, Simpson indices, and vegetative variables analyzed for their contributions to the first two principal components. PCA was done to find the association of altitudes with reduced biomass production, lower carbon sequestration, and poorer species diversity—likely attributable to harsher environmental conditions, such as decreased temperatures, limited soil nutrients, and elevated climatic stresses. For the analysis and graphical visualization, R-studio v.4.4.2, Origin Pro software (v.2022) and SPSS software (IBM, New York, NY, USA, version 20) were used.

## 5. Conclusions

This study demonstrates that elevation gradients significantly influence conifer forest structure, carbon storage, species diversity, and regeneration capacity in the Kalam temperate forests of Pakistan. While tree height, DBH, and basal area increased with elevation, carbon stock, species density, and regeneration potential declined. Structural attributes such as volume and basal area were the strongest predictors of carbon stock, while species identity, particularly *Cedrus deodara* and *Abies pindrow*, shaped regeneration and carbon patterns across altitudes. Litter and deadwood carbon, along with SOCs, varied with elevation, peaking at mid-elevations, suggesting complex interactions between biotic and abiotic factors. Lower elevations experienced higher deforestation, reducing carbon storage potential, while higher altitudes were constrained by climatic and edaphic limitations. PCA analysis confirmed elevation as a major driver of ecological variability. These findings offer critical insights into altitudinal responses of forest ecosystems and highlight the importance of elevation-sensitive management for biodiversity conservation and climate change mitigation in temperate coniferous forests.

## Figures and Tables

**Figure 1 plants-14-01534-f001:**
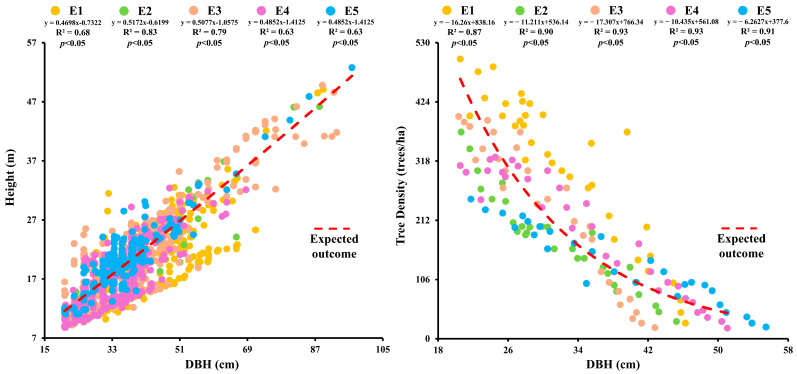
Relationships of tree DBH with tree height and stand density at different elevations in the forest of Kalam valley, Pakistan.

**Figure 2 plants-14-01534-f002:**
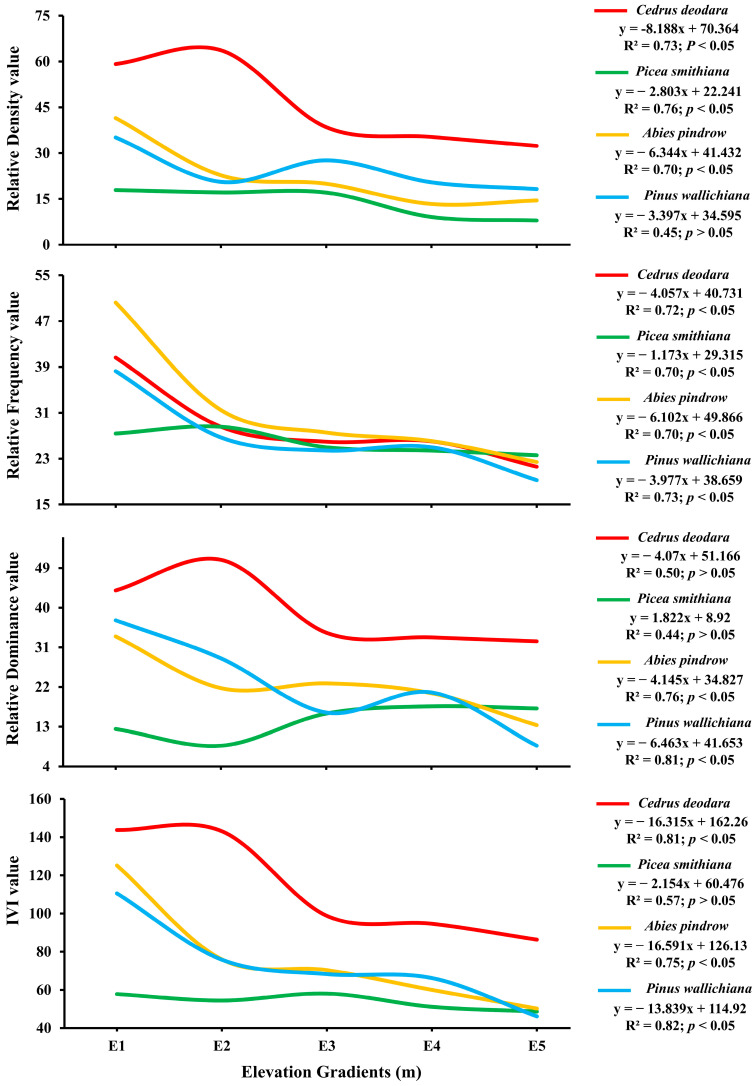
Influence of elevation gradients on four conifer tree species based on Relative Density (RD), Relative Abundance (RA), Relative Frequency (RF) and Important Value Index (IVI) at significance level (*p* < 0.05) in the forest of Kalam valley, Pakistan.

**Figure 3 plants-14-01534-f003:**
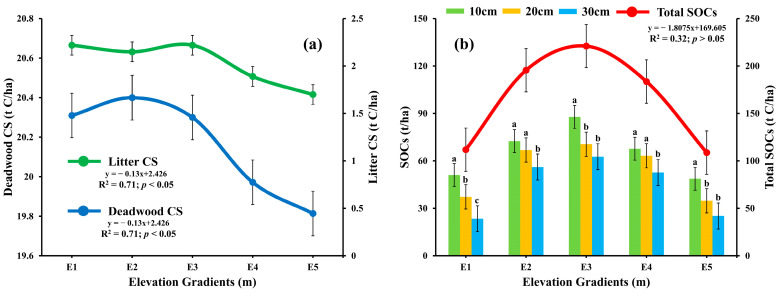
Influence of elevation gradients on (**a**) deadwood, and litter C stocks, and (**b**) SOCs at three different depths in the conifer forest of Kalam valley, Pakistan. Total SOCs is the sum of SOCs from all three depths. Letters (a, b, c) indicate significant differences among the SOCs C stocks at each elevation based on Tukey’s post Hock test (*p* ≤ 0.05).

**Figure 4 plants-14-01534-f004:**
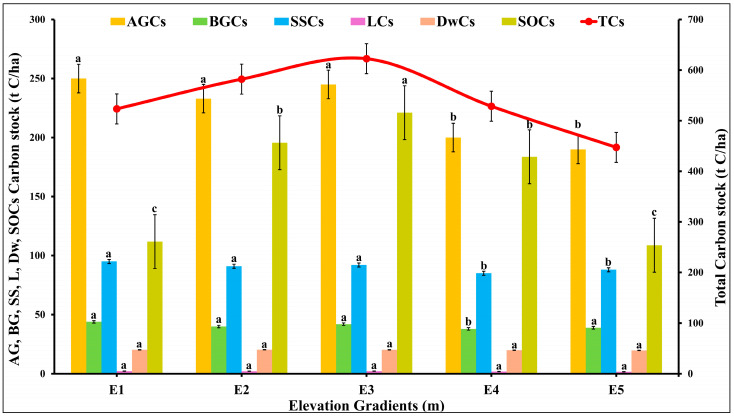
Influence of elevation gradients on the C pool in the forest of Kalam valley, Pakistan. AGCs = Above ground carbon stock, BGCs = Below ground carbon stock, SSCs = Standing sapling carbon stock, LCs = Litter carbon stock, DwCs = Deadwood carbon stock, SOCs = Soil organic carbon stock, and TCs = Total carbon stock. Letters (a, b, c) indicate the significance different between the mean value of each variable for all five elevation gradients at significance level (*p* < 0.05) (based on Tukey post Hock test). Error bars are indicating the standard error.

**Figure 5 plants-14-01534-f005:**
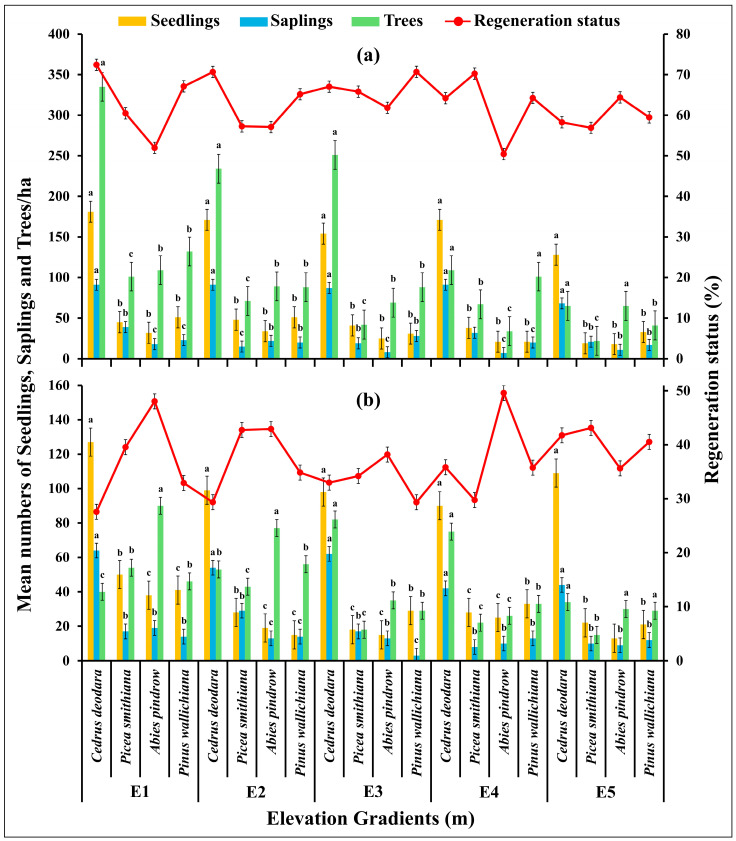
The regeneration ability of (**a**) low disturbed and (**b**) high disturbed forest areas varied at different elevation gradients in Kalam valley, Pakistan. Error bars indicate the standard errors. Letters (a, b, c) indicate the significance different between the species groups at each elevation gradient for each variable at *p* < 0.05, based on Tukey post Hock test.

**Figure 6 plants-14-01534-f006:**
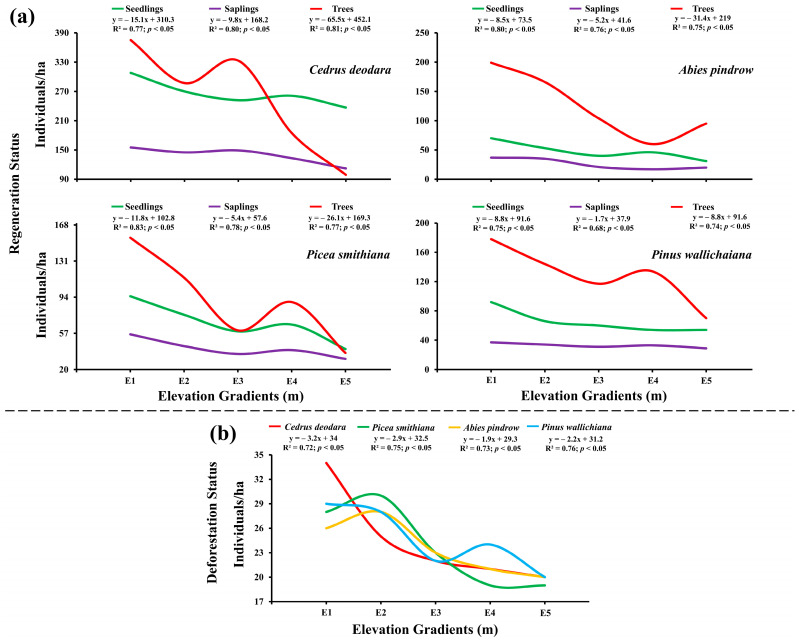
Influence of elevational gradients on (**a**) regeneration status and (**b**) deforestation status of *Cedrus deodara*, *Picea smithiana*, *Abies pindrow*, and *Pinus wallichiana* in the forest of Kalam valley, Pakistan.

**Figure 7 plants-14-01534-f007:**
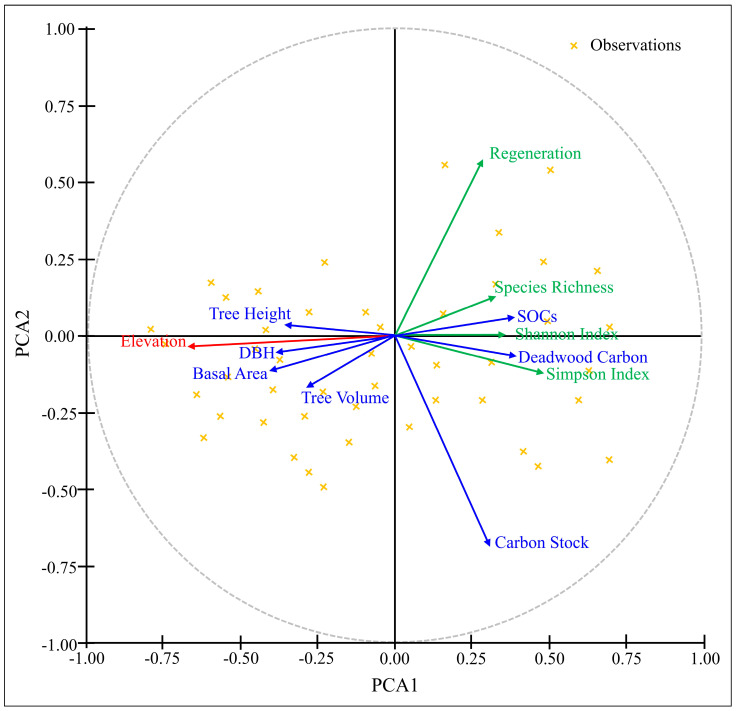
Relationships among environmental factor elevation with vegetation attributes i.e., DBH, height, basal area, volume, carbon stock, SOC, regeneration, Shannon index, Simpson index and species richness based on PCA.

**Figure 8 plants-14-01534-f008:**
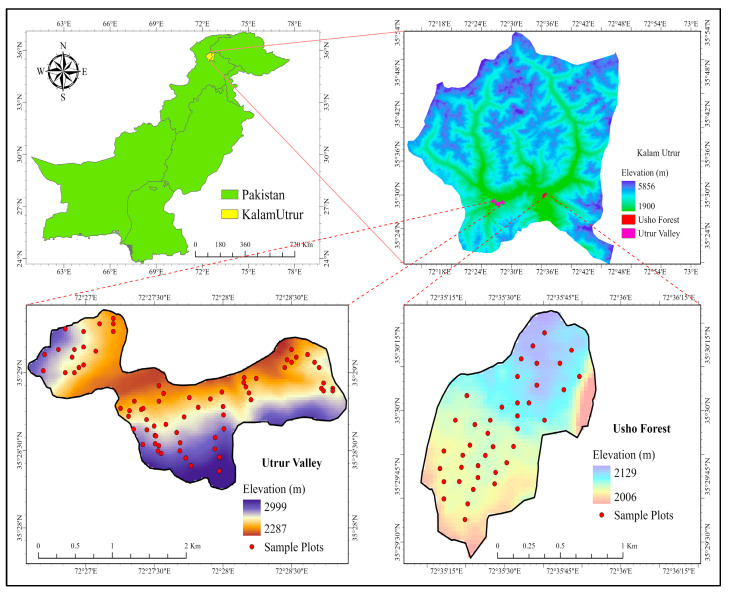
The study area Kalam valley, Swat district, Pakistan with the sampling plots.

**Figure 9 plants-14-01534-f009:**
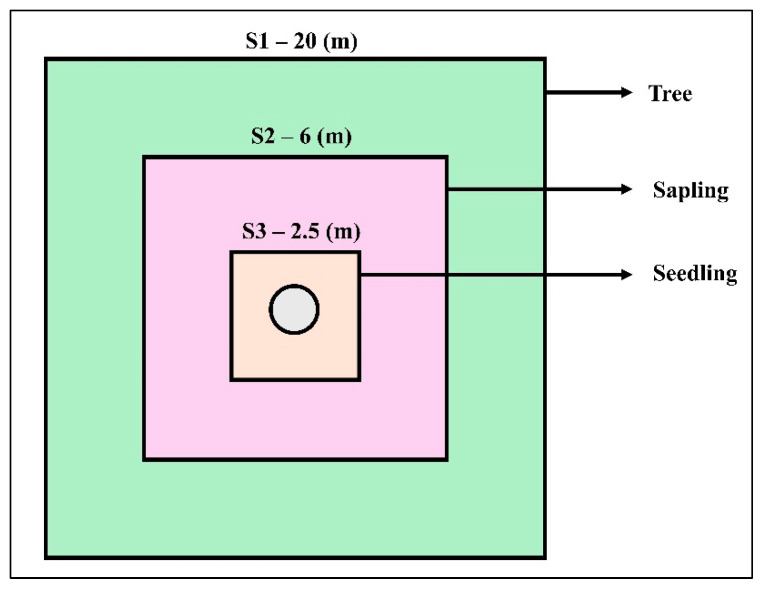
Nested plot sampling design for trees, saplings, and seedlings at Kalam valley, Swat district, Pakistan.

**Table 1 plants-14-01534-t001:** Multiple linear regression coefficients showing the influence of species, topographic, site, and stand structural variables on tree carbon stock (TCS).

Variable	Categories	B	Std. Error	Beta	t	Sig.	Tolerance	VIF
Location	Usho	−9.456	2.82	−0.035	−3.354	<0.001 ***	0.175	5.704
Slope	Gentle	−2.176	2.345	−0.008	−0.928	0.354 ^NS^	0.235	4.251
	Moderate	−3.37	2.612	−0.009	−1.29	0.197 ^NS^	0.391	2.555
	Steep	1.978	2.653	0.008	0.746	0.456 ^NS^	0.168	5.948
Aspect	G2	−3.387	1.313	−0.013	−2.579	0.01 **	0.819	1.22
	G3	−4.484	1.595	−0.014	−2.812	0.005 **	0.831	1.204
Species	Fir	−65.847	1.531	−0.212	−43.021	<0.001 ***	0.805	1.242
	Sp	−43.018	1.699	−0.124	−25.313	<0.001 ***	0.815	1.227
	BP	−41.91	1.454	−0.142	−28.829	<0.001 ***	0.803	1.245
Elevation	E2	−6.943	2.484	−0.014	−2.796	0.005 **	0.776	1.288
	E3	−7.447	2.786	−0.029	−2.673	0.008 **	0.167	5.995
	E4	−2.015	2.931	−0.007	−0.687	0.492 ^NS^	0.19	5.267
	E5	−7.388	3.236	−0.016	−2.283	0.023 *	0.383	2.609
Basel area		59.148	2.116	0.409	27.947	<0.001 ***	0.091	10.946
Volume		2.06	0.05	0.6	41.607	<0.001 ***	0.094	10.656

B is the regression coefficient; beta is the standardized regression coefficient. Significant predictors (*p* < 0.05) are highlighted, with Diar, E1, Utror, Plane slope, and G1 aspect serving as reference categories. Continuous variables included basal area and tree volume. VIF values were used to assess multicollinearity. G2 = South/South-East/South-West, G3 = East/West, Fir = *Abies pindrow*, Sp = *Picea smithiana*, BP = *Pinus wallichiana*, NS = non-significant, * = significant at *p* < 0.05, ** = significant at *p* < 0.01, and *** = significant at *p* < 0.001.

**Table 2 plants-14-01534-t002:** Association between elevation gradients, species diversity, evenness, and richness in the forest of Kalam valley, Pakistan. Letters (^a, b^) indicate significant differences among the indices at each elevation (*p* ≤ 0.05).

Elevation	Shannon Index H	Simpson Index D	Species Evenness E	Species Richness S
**E1**	1.18 ± 0.01 ^a^	3.28 ± 0.04 ^b^	0.91 ± 0.01 ^a^	3.68 ± 0.02 ^a^
**E2**	1.15 ± 0.02 ^a^	3.18 ± 0.03 ^a^	0.91 ± 0.01 ^a^	3.67 ± 0.02 ^a^
**E3**	1.09 ± 0.01 ^b^	2.98 ± 0.02 ^b^	0.89 ± 0.01 ^a^	3.67 ± 0.02 ^a^
**E4**	1.10 ± 0.02 ^a^	3.07 ± 0.03 ^a^	0.87 ± 0.01 ^b^	3.66 ± 0.02 ^a^
**E5**	1.09 ± 0.01 ^b^	2.94 ± 0.02 ^a^	0.87 ± 0.01 ^b^	3.67 ± 0.02 ^a^
**Adj. R^2^**	0.72	0.73	0.87	0.16
**F**	11.41	11.15	27.00	2.45
***p*-value**	0.04	0.04	0.01	0.22

**Table 3 plants-14-01534-t003:** Climatic features of the study area.

Month	Average Temperature (°C)	Minimum Temperature (°C)	Maximum Temperature (°C)	Precipitation (mm)	Relative Humidity (%)	Rainy Days
January	−13.7 ± 4.32	−17.2 ± 6.12	−9.8 ± 2.12	111 ± 11.22	56 ± 6.32	9 ± 1.61
February	−11.2 ± 3.92	−15 ± 5.32	−7.3 ± 1.31	166 ± 16.83	61 ± 6.46	10 ± 2.01
March	−7.1 ± 3.11	−11.5 ± 3.44	−2.7 ± 0.44	196 ± 20.12	60 ± 6.16	11 ± 2.21
April	−2.4 ± 1.13	−7.3 ± 1.92	1.9 ± 0.24	168 ± 17.32	59 ± 5.91	10 ± 2.11
May	2.5 ± 1.23	−3.1 ± 1.02	7.2 ± 1.04	102 ± 10.72	63 ± 6.56	8 ± 1.31
June	8.1 ± 2.12	1.3 ± 0.32	13.3 ± 3.12	88 ± 9.12	64 ± 6.22	7 ± 1.01
July	12 ± 3.78	6.1 ± 2.67	16.9 ± 5.02	127 ± 12.52	74 ± 6.13	17 ± 2.16
August	11.5 ± 3.32	6.1 ± 2.71	16.2 ± 4.98	110 ± 11.32	77 ± 7.01	12 ± 1.98
September	7.4 ± 3.32	1.4 ± 0.35	12.6 ± 3.12	78 ± 9.99	71 ± 7.12	7 ± 1.01
October	−0.1 ± 0.12	−6.3 ± 3.31	5.6 ± 2.92	68 ± 9.78	64 ± 6.16	8 ± 1.21
November	−7.9 ± 1.22	−12.5 ± 4.62	−3.2 ± 1.12	70 ± 10.22	59 ± 5.96	6 ± 0.98
December	−12.5 ± 35	−16.1 ± 5.22	−8.6 ± 3.02	77 ± 10.32	55 ± 5.11	6 ± 0.97

Source: https://en.climate-data.org/.

**Table 4 plants-14-01534-t004:** Allometric equations used to calculate the above-ground biomass of four conifer species.

Species	Equation	Reference
*Cedrus deodara*	$AGB=0.1779\times {\left(D^{2}\times H\right)}^{0.8103}$	[74]
*Pinus wallichiana*	$AGB=0.0594\times {\left(D^{2}\times H\right)}^{0.881}$	[75]
*Picea smithiana*	$AGB=0.0821\times {\left(D^{2}\times H\right)}^{0.8363}$	[76]
*Abies pindrow*	$AGB=0.0452\times {\left(D^{2}\times H\right)}^{0.9029}$	[77]

## Data Availability

All the data is presented in the manuscript.

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
