# Peer review of "Altitudinal Variations in Coniferous Vegetation and Soil Carbon Storage in Kalam Temperate Forest, Pakistan"

_plants, 2025, doi:10.3390/plants14101534_

Round 1
Reviewer 1 Report
Comments and Suggestions for Authors
The paper being a relatively simple empirical study, it apparently does not contain fundamental flaws. However, there appears to be much to be corrected.
Abbreviations are used without explaining their meaning-
Climate and weather characteristics are given without clarification of their context.
“Total C stock ” is given without SOC.
Equations contain undefined symbols.
The definition of disturbance levels is unclear to this reviewer.
Imperial and metric measures are mixed.
Figure captions refer to elevation gradients, even if there are only elevations given in the Figures.
In Fig. 7, the same symbols are used in two different meanings.
In Table 2, units are missing
The Discussion appears to contain self-contradictory statements.
Author Response
- Q. Abbreviations are used without explaining their meaning-Climate and weather characteristics are given without clarification of their context.
Answer: Thanks, you for your comments we have addressed the comment in the revised manuscript.
- Q. “Total C stock ” is given without SOC.
Answer: Thank you for your comment, we have revised the equation also revised the figure, analysis and modify the methodology and result sections.
- Q. Equations contain undefined symbols.
Answer: All the equations have been revised.
- Q. The definition of disturbance levels is unclear to this reviewer.
Answer: Thank you for your comment. We have clarified the definition of disturbance levels in the revised manuscript to ensure better understanding. The criteria for categorizing disturbance levels are now more explicitly defined based on proximity to human settlements, grazing intensity, and deforestation rates.
Revised Text in Manuscript: Disturbance levels were classified based on anthropogenic pressure indicators such as proximity to human settlements, intensity of grazing, and extent of deforestation. Forest areas between 2000–2400 m was categorized as high disturbance zones due to their close proximity to communities and nomadic territories, characterized by frequent livestock grazing and significant deforestation. In contrast, areas above 2400 m were defined as low disturbance zones, as they were more remote, experienced minimal human presence, low grazing pressure, and limited deforestation.
- Q. Figure captions refer to elevation gradients, even if there are only elevations given in the Figures.
Answer: Thank you so much for highlighting it, based on your comment we have revised the whole result section thoroughly. You can find the major changes in the methodology and result sections in the revised MS.
- Q. In Fig. 7, the same symbols are used in two different meanings.
Answer: All figures and result section have been revised thoroughly. For detail please see the revised MS.
- Q. In Table 2, units are missing
Answer: All the tables have been revised.
- Q. The Discussion appears to contain self-contradictory statements.
Answer: Revised.
Reviewer 2 Report
Comments and Suggestions for Authors
The manuscript claims to deal with the coniferous vegetation and soil carbon storage dynamics across altitudinal gradients in Kalam temperate forest, Pakistan. It is based on an interesting data base, and mainly deals with the dependence of carbon storage on elevation.
A main flaw in this manuscript is the way how it deals with units. In parts the authors use Anglican units (inches and feet) and in other parts the use metric units (cm, m,..). In some parts they don’t give any units are wrong ones. I think in an international scientific journal only metric units should be used.
Examples:
Equations (1) to (4): Without the units given, the equations are useless. Why do these equtions have two factors, they could be combined into one factor, or should the last factor be meant as an exponent? In line 206 the authors say that “below ground biomass and total tree biomass were calculated after doing the above calculations. The question is how these were calculated!
In Table 2 they give the number of seedlings, saplings and trees. Is this per hectare or per plot.
In Figure 1, the dbh is given in inches, instead of cm. an in Table S2, The dbh is said to be given in meters, which cannot be correct, because dbh around 13-20 m are not plausible.
A second problem is with the statistical calculations:
Lines 301 and in other places the R² is negative. Negative R² cannot be, because it is a square! I think the authors put the minus sign to the R² when the dependent variable decreased with the independent one. Nevertheless, this is wrong!
In line 254, the authors say that” t-tests were used to check the significant data with respect to elevation for each species and to check the significance within the groups”. In fact later on they tested the differences by Tukey post-hoc tests – which is correct, while the use of t-tests for the differences in these cases is incorrect. Since they do not say what these “groups” were, it is very difficult to understand what differences within “groups” are.
The multiple linear regression (Table 1) with categorical variables is not clear. It is important to know how the categories were coded. Obviously they did not use the concept of dummy variables, because otherwise the cannot be only one regression coefficient for the variable. Furthermore the authors seem not to have tested for inter-correlations, because dbh and basal area must be highly correlated so that they should not be used both in the same regression.
One more problem are the equations in lines 221-223. The equation for the Shannon index is correct. In line 217 the authors say that they calculated Pielou’s evenness, but J´ is the evenness of the Shannon index, but the equation for H´max is wrong, it should be simply the natural logarithm of the number of species. The Simpson-diversity index is 1-sum(pi²), the three equations under (9) are wrong. This is the reason why later on they find a negative correlation of the Shannon index with the elevation why the correlation with the Simpson index is positively correlated with the elevation. Since both indices indicate diversity, they both should be correlated with the elevation in the same direction.
The equations (10)-(12) need a definition of frequency, because otherwise in normal understanding (10) = and (12) should be equal. These definitions are also needed for Table S3. Otherwise one cannot understand why relative density, frequency, and abundance are different.
Finally, there are many typos in the manuscript which would have been easily detected by a spell checker.
Examples: Line 86: altididinal; line 94: biomss¸ line 391: salings; line 404 and 406: negtive, line 475 eisting, line 531: significalty, line 537: elevtion.
Please understand that these are only examples, I did not feel obliged to edit the whole manuscript.
Comments on the Quality of English LanguageThe English is not so bad, but there are too much typos in the manuscript.
Author Response
- Q. A main flaw in this manuscript is the way how it deals with units. In parts the authors use Anglican units (inches and feet) and in other parts the use metric units (cm, m,..). In some parts they don’t give any units are wrong ones. I think in an international scientific journal only metric units should be used. Examples: Equations (1) to (4): Without the units given, the equations are useless. Why do these equtions have two factors, they could be combined into one factor, or should the last factor be meant as an exponent? In line 206 the authors say that “below ground biomass and total tree biomass were calculated after doing the above calculations. The question is how these were calculated!
Answer: Thank you for your comment. We have revised all the equations, also added the missing information and modified the units to metric units as suggested. For details please see the revised MS.
- Q. In Table 2 they give the number of seedlings, saplings and trees. Is this per hectare or per plot.
Answer: Thank you for your comment. We have revised the units and added the units for all those that did have units before. However, the number of seedlings, saplings, and trees were calculated per hectare.
- Q. In Figure 1, the dbh is given in inches, instead of cm. an in Table S2, the dbh is said to be given in meters, which cannot be correct, because dbh around 13-20 m are not plausible.
Answer: Thank you for highlighting it. We have revised the figures and modified the units as metric units (as suggested before), also we modified the values in table S2. Now the values are in cm in table S2.
- Q. Lines 301 and in other places the R² is negative. Negative R² cannot be, because it is a square! I think the authors put the minus sign to the R² when the dependent variable decreased with the independent one. Nevertheless, this is wrong!
Answer: Thank you for your valuable comment. We have carefully reviewed all instances where R² values were reported and have removed any incorrect negative signs. We acknowledge that R², being the square of the correlation coefficient (r), cannot be negative and should always be reported as a positive value. The inverse or negative relationship between variables has now been correctly described using the correlation coefficient (r) where appropriate, rather than misrepresenting R². These corrections have been made in both the text and figures to ensure accuracy.
- Q. In line 254, the authors say that” t-tests were used to check the significant data with respect to elevation for each species and to check the significance within the groups”. In fact later on they tested the differences by Tukey post-hoc tests – which is correct, while the use of t-tests for the differences in these cases is incorrect. Since they do not say what these “groups” were, it is very difficult to understand what differences within “groups” are.
Answer: Thank you for your valuable comment. We agree with your observation and have reviewed the methods section accordingly. The mention of the t-test was a typographical error. As correctly pointed out, Tukey’s post-hoc test was used to assess the differences among groups based on elevation for each species. This has now been clearly corrected and clarified in the revised manuscript.
- Q. The multiple linear regression (Table 1) with categorical variables is not clear. It is important to know how the categories were coded. Obviously they did not use the concept of dummy variables, because otherwise the cannot be only one regression coefficient for the variable.
Answer: Thank you for your insightful comment. We acknowledge the importance of properly representing categorical variables in multiple linear regression models. In our original analysis, categorical variables such as location (2 categories), elevation (5 categories), aspect (3 categories), and slope (4 categories) were indeed included. However, we realize now that the manuscript did not clearly explain how these categories were treated.
We confirm that dummy variables were not used in the model presented in Table 1, and instead, these categorical variables were incorrectly treated as continuous predictors, which can misrepresent their effect. We have now revised the MLR by modifying the factorial variable and by properly converting categorical variables into dummy variables before re-running the regression analysis. In the revised analysis we also added the species factor. We categories the variables as follows.
Variable |
Categories |
Dummies |
Reference |
Location |
Utror – Usho |
Usho |
Utror |
Elevation |
E1 – E5 |
E2, E3, E4, E5 |
E1 |
Slope |
Plane, Gentlr, Moderate, Steep |
Gentle, Moderate, Steep |
Plane |
Aspect |
G1 (N/NE/NW), G2 (S/SE/SW), G3 (E/W) |
G2, G3 |
G1 |
Species |
Diar – Cedrus deodara Fir – Abies pindrow Sp – Picea smithiana BP – Pinus wallichiana |
Fir, Sp, BP |
Diar |
The revised model includes appropriate coding for each categorical predictor, and we have updated the results and description in the manuscript accordingly. This correction improves the interpretation of regression coefficients and ensures the validity of the model assumptions. We thank the reviewer for bringing this to our attention.
- Q. Furthermore the authors seem not to have tested for inter-correlations, because dbh and basal area must be highly correlated so that they should not be used both in the same regression.
Answer: We have revised the whole MLR analysis. Please find the details in revised MS.
Q1 . One more problem are the equations in lines 221-223. The equation for the Shannon index is correct. In line 217 the authors say that they calculated Pielou’s evenness, but J´ is the evenness of the Shannon index, but the equation for H´max is wrong, it should be simply the natural logarithm of the number of species. The Simpson-diversity index is 1-sum(pi²), the three equations under (9) are wrong. This is the reason why later on they find a negative correlation of the Shannon index with the elevation why the correlation with the Simpson index is positively correlated with the elevation. Since both indices indicate diversity, they both should be correlated with the elevation in the same direction.
Answer: Thank you for the comment and highlighting the mistake. As you instructed, we have revised the formula and re calculated the values and run the analysis again, also modified the figures accordingly. Please see the details in the revise manuscript.
- Q. The equations (10)-(12) need a definition of frequency, because otherwise in normal understanding (10) = and (12) should be equal. These definitions are also needed for Table S3. Otherwise one cannot understand why relative density, frequency, and abundance are different.
Answer: Thank you for your comment, for more clarification we have added the definitions.
Where Frequency of a species is defined as the number of sampling units in which a particular species occurs, and Sum of all frequencies is defined as the total number of sampling units in which all species combined occur (i.e., the sum of frequencies for each species).
- Q. Finally, there are many typos in the manuscript which would have been easily detected by a spell checker.
Examples: Line 86: altididinal; line 94: biomss¸ line 391: salings; line 404 and 406: negtive, line 475 eisting, line 531: significalty, line 537: elevtion.
Please understand that these are only examples, I did not feel obliged to edit the whole manuscript.
The English is not so bad, but there are too much typos in the manuscript.
Answer: We have revised the whole MS accordingly to address this issue.
Reviewer 3 Report
Comments and Suggestions for Authors
General comments
This is a highly descriptive study and describes the state of forests at various elevations. Many of the analyses are correlational in nature and for the most part reveal little that is new or of general importance.
The manuscript is repetitive with many of the ideas occurring more than once. There is a lack of precision in terms of descriptions of relationships, cause and effect, mechanisms, and processes. For example, the word dynamic is used repeatedly, but dynamics, that is the change over time, was not studied. Some of the relationships found amount to circular logic. For example, correlating variables used to calculate biomass and carbon stores with biomass or carbon stores makes little sense. I also found the manuscript to be poorly organized making it difficult to follow the logic. Part of the problem is that so many variables were examined that a focus was hard to determine.
Specific comments (line)
2 This study does not study the dynamics of any of the variables examined. It does examine the state of various variables. The title needs to be changed.
27 The authors did not study soil dynamics.
31 how large were the plots?
34 In a mechanistic sense elevation does not influence plants or soils directly. What does effect plants and soils directly is temperature, moisture, light, etc.
40 One cannot assess dynamics in a one-time sampling scheme. Dynamics implies change over time.
56 This involves circular logic and does not tell the reader anything of value given that trees are of different species and variation in tree diversity must be related to species diversity.
60 But do these really influence the vegetation. I understand that they are related to changes in vegetation, but they are not direct causes of change. That would involve, water, light, nutrients, and temperature.
77 Isn’t this just repeating material from the first paragraph?
100 Richness is an index of diversity. It is therefore hard to understand what this means. Are the author referring to other indices of diversity such as evenness and heterogeneity?
104 Several suggests just a few, but there are hundreds of studies on this topic.
110 unless this study involves measuring processes or change in states over time it can not possibly provide any insights into dynamics.
130 The hypothesis is so general it cannot be falsified. Indicating something changes with elevation does not create a falsifiable hypothesis because it would be true if there is an increase or a decrease.
153 Given that most soils can support some form of vegetation it is not clear what this statement adds.
156 It is not clear how the dominate species, which mostly seem to be evergreen are deciduous as suggested by the previous sentence.
159-162 This material would be better in the introduction.
209 How was the dead wood determined? What measurements were used?
219 It is not clear which index is being calculated by this formula.
236 By density, is bulk density what is meant? Or is it particle density? I am not sure how one could determine the bulk density from an augured sample. One needs to know the volume. How was this determined?
278 temperate is misspelled.
280 given that tree DBH and height are used to determine biomass, why would anyone include those variables in a regression seeking uncover relationships? Isn’t this already known?
305 as written this seems to suggest that there is a correlation between basal area and basal area.
342-344 This is methods and not results.
351 The differences between the elevation zones do not seem particularly large. However, since there is no indication of uncertainty or variability it would be impossible to judge whether this is the case.
410-414 There is an increase in font size in this section of text.
421 It is not clear what beyond the forest parameter range means.
428-438 This is largely discussion material and not results. Interpretations of findings are to be found in the discussion section.
446 Most of these variables would have to be correlated with biomass and carbon stocks because they are either correlated with or part of the biomass calculation. This sort of analysis provides no actual insights.
454 If trees are growing at all elevation levels, then aren’t there sufficient nutrients at all elevation levels to support trees? it is not clear what is meant as written.
Comments on the Quality of English Language
see specific comments
Author Response
2 This study does not study the dynamics of any of the variables examined. It does examine the state of various variables. The title needs to be changed.
Answer: Copy that. Revised and modified.
27 The authors did not study soil dynamics.
Answer: Copy that. Revised and modified.
31 how large were the plots?
Answer: Each plot was 20×20m (400m2). This information has been added in the abstract.
34 In a mechanistic sense elevation does not influence plants or soils directly. What does effect plants and soils directly is temperature, moisture, light, etc.
Answer: Copy that. We have revised and modified the passage to address any uncertainty.
40 One cannot assess dynamics in a one-time sampling scheme. Dynamics implies change over time.
Answer: Copy that. We have revised and modified the passage to address any uncertainty.
56 This involves circular logic and does not tell the reader anything of value given that trees are of different species and variation in tree diversity must be related to species diversity.
Answer: We have revised and modified the section to address any uncertainty.
60 But do these really influence the vegetation. I understand that they are related to changes in vegetation, but they are not direct causes of change. That would involve, water, light, nutrients, and temperature.
Answer: We have revised and modified the passage to address any uncertainty.
77 Isn’t this just repeating material from the first paragraph?
Answer: Copy that. We have revised and modified the passage to address any uncertainty.
100 Richness is an index of diversity. It is therefore hard to understand what this means. Are the author referring to other indices of diversity such as evenness and heterogeneity?
Answer: We have revised and modified the passage to address any uncertainty.
110 unless this study involves measuring processes or change in states over time it can not possibly provide any insights into dynamics.
Answer: Copy that. We have revised and modified the passage to address any uncertainty.
130 The hypothesis is so general it cannot be falsified. Indicating something changes with elevation does not create a falsifiable hypothesis because it would be true if there is an increase or a decrease.
Answer: Thank you for your valuable observation. We agree that the original hypothesis was broad and lacked falsifiability. In response, we have revised the hypothesis to be more specific, directional, and testable. The revised hypothesis now reads: “We hypothesize that increasing elevation is associated with a decline in the growth rate and carbon stock of conifer tree species due to environmental constraints such as lower temperatures, reduced soil fertility, and shorter growing seasons. Consequently, higher elevations are expected to support reduced tree development, biodiversity, and soil carbon storage compared to lower elevations.” This revised formulation makes the hypothesis directional and falsifiable by clearly stating the expected relationship, which can be tested using our regression models and field data. We have updated the manuscript accordingly in the Introduction section.
153 Given that most soils can support some form of vegetation it is not clear what this statement adds.
Answer: Thank you for your comment. We have revised and modified the passage to address any uncertainty.
156 It is not clear how the dominate species, which mostly seem to be evergreen are deciduous as suggested by the previous sentence.
Answer: We apologize for the mistake. It was a typo. Thank you for highlighting it. It has been corrected.
159-162 This material would be better in the introduction.
Answer: Moved to the introduction section as suggested.
209 How was the dead wood determined? What measurements were used?
Answer: Thank you for highlighting it. To address this question, we have added the brief passage in a methodology section. For detail, please see the revised MS.
219 It is not clear which index is being calculated by this formula.
Answer: Based on your comment we have revised the formula and now clearly mention which index is being calculated by which equation. For detail please see the methodology section of revised MS.
236 By density, is bulk density what is meant? Or is it particle density? I am not sure how one could determine the bulk density from an augured sample. One needs to know the volume. How was this determined?
Answer: To address this question we have added a detailed passage along with references in the methodology section of the revised MS.
278 temperate is misspelled.
Answer: Corrected.
280 given that tree DBH and height are used to determine biomass, why would anyone include those variables in a regression seeking uncover relationships? Isn’t this already known?
Answer: We have revised the regression model (MLR) analysis for detail please see the revised MS.
305 as written this seems to suggest that there is a correlation between basal area and basal area.
Answer: Corrected.
342-344 This is methods and not results.
Answer: As suggested, we have removed the whole passage from the result section.
351 The differences between the elevation zones do not seem particularly large. However, since there is no indication of uncertainty or variability it would be impossible to judge whether this is the case.
Answer: We have revised the analysis and reconstructed the figures, now the figures and description are much clearer and more scientific. For detail, please see the result section of revised MS.
410-414 There is an increase in font size in this section of text.
Answer: Checked and corrected.
421 It is not clear what beyond the forest parameter range means.
Answer: Revised as suggested.
428-438 This is largely discussion material and not results. Interpretations of findings are to be found in the discussion section.
Answer: Copy that, revised and modified.
446 Most of these variables would have to be correlated with biomass and carbon stocks because they are either correlated with or part of the biomass calculation. This sort of analysis provides no actual insights.
Answer: We have revised the PCA analysis to better address this problem.
454 If trees are growing at all elevation levels, then aren’t there sufficient nutrients at all elevation levels to support trees? it is not clear what is meant as written.
Answer: Whole passage has been revised and modified.
Round 2
Reviewer 2 Report
Comments and Suggestions for Authors
The authors obviously struggled to improve the manuscript, guided by my comments in the first review. A first comment now is that the corrected manuscript contains all the corrections in red, which makes it very difficult to read. They should have used the view which without these correction before converting the word file into a pdf.
Unfortunately the new corrections are quite incomplete. Maybe they did not use the equation editor or they did not fully understand my comments.
In the equations in Table 1, the last number must be a exponent and not a factor, thus the first equation should run as: AGB = 0.1779Ë‘(D² Ë‘H)0.8103
In equation (4) the p is missing, obviously the volume of cone is supposed.
The Simpson index in equation (9) is still wrong. D=1-D is impossible, and probably the numerator and the denominator are interchanged. Please once more have a look on what you have calculated, the Shannon index and the Simpson index cannot be negatively correlated!
Positive is that the authors now used dummy variables for Species, slope, and elevation.
A problem is still the coefficient of determination and the correlation coefficient. In some places, the authors kept the number of the coefficient and simply changed R² to r. Since I do not have the data I cannot check what they really calculated. From Figures 3 and 4 and S1 I conclude that they calculated R². Then the r is the square root of R² and not the same number (see as an example line 40).
The term gradients seem to have been misunderstood. (see line 32 as an example): There is one gradient, consisting of the five altitudinal zones, and not 5 gradients.
Because there ae still errors in the methods section it does not make sense to discuss the results and the conclusions, because the results may be different if correctly calculated.
Author Response
The authors obviously struggled to improve the manuscript, guided by my comments in the first review. A first comment now is that the corrected manuscript contains all the corrections in red, which makes it very difficult to read. They should have used the view which without these correction before converting the word file into a pdf.
Answer: Sorry for last time, this time we will provide the clean copy of revised MS as a pdf file.
Unfortunately the new corrections are quite incomplete. Maybe they did not use the equation editor or they did not fully understand my comments. In the equations in Table 1, the last number must be a exponent and not a factor, thus the first equation should run as: AGB = 0.1779Ë‘(D² Ë‘H)0.8103
Answer: Thank you for your comments. We have revise the equations.
The Simpson index in equation (9) is still wrong. D=1-D is impossible, and probably the numerator and the denominator are interchanged. Please once more have a look on what you have calculated, the Shannon index and the Simpson index cannot be negatively correlated!
Answer: Thank you for your comments, as you suggested we have revised the Simpson index of diversity with the corrected and recalculated the index values. Please find the correction in the revised MS. We appreciate the reviewer’s valuable comments regarding the relationship between Shannon and Simpson indices. In our study, both indices showed a positive correlation across elevations, which is consistent with ecological theory: when Shannon diversity increases, Simpson diversity also increases. However, our results specifically indicated that lower elevations (E1, E2) exhibited significantly higher diversity and evenness compared to higher elevations (E3, E4, E5). Therefore, there is no contradiction, and the positive association between Shannon and Simpson indices was maintained throughout the analysis.
A problem is still the coefficient of determination and the correlation coefficient. In some places, the authors kept the number of the coefficient and simply changed R² to r. Since I do not have the data I cannot check what they really calculated. From Figures 3 and 4 and S1 I conclude that they calculated R². Then the r is the square root of R² and not the same number (see as an example line 40).
Answer: As you mentioned that the R2 can not be negative and obviously it is true, so we have revised the coefficient values and changed them from – to simple values. We genuinely calculated the regression values and used the adjusted R2 values for presentation.
The term gradients seem to have been misunderstood. (see line 32 as an example): There is one gradient, consisting of the five altitudinal zones, and not 5 gradients.
Answer: Understood. Corrected.
Because there ae still errors in the methods section it does not make sense to discuss the results and the conclusions, because the results may be different if correctly calculated.
Answer: We have revised the MS thoroughly. We hope you will appreciate the corrections this time and support MS for publication.
Reviewer 3 Report
Comments and Suggestions for Authors
General comments
While the revisions of the authors have improved the manuscript and in some cases provided much needed detail to understand the study, the manuscript is still problematic as written. The main points of concern:
1 It is still hard to find a logical flow in the introduction. Points appear over and over in different paragraphs.
2 The justification for the study also remains weak. It seems to be that because these relationships were not studied in this particular location that it needs to be studied. That may be, but that is largely a local and not a global justification.
3 The discussion is not compelling nor is it logically arranged. Again as with the introduction points seem to be repeated in different paragraphs. And the importance and significance of the findings is not made clear.
The methods described seem to only deal with standing dead wood. How was downed dead wood measured or was it excluded? If excluded the authors need to make sure their terminology regarding results reflects that.
Specific comments (line)
63-65 The meaning of this sentence is not clear. What declines? The species? The peak performance? How can a peak performance decline as suggested? Also, how do species diversity and vegetation diversity differ? How are they different?
112 richness is an index of biodiversity. Biodiversity can be described using richness, evenness, and heterogeneity. So this statement does not make sense as written.
149-150 This is a study objective. It should therefore go with the other study objective. It would also be helpful to have a hypothesis related to this objective so that it matches the other study objective. Given the lack of background on this topic in the introduction it would also make sense to have at least one paragraph on this topic in the introduction. Otherwise it seems likes something that was just added on.
279 It is not clear how this method could determine soil bulk density. It could certainly be used to determine the mass in a certain size fraction.
300 It is not clear what “deliberated” means in this context. Do the authors mean “determined”?
599-600 This sentence is not needed. It should be placed in the methods to justify the use of PCA.
627 The correlations between carbon stocks and basal area, volume, height and DBH do not anything to do with the things listed in the next sentence. They have to do with allometric relationships.
635 As noted before, if trees are present, then if must be that there is sufficient resources for tree growth. The authors must be referring to something else, but as written this is circular logic.
647 It would have been more interesting to contrast the different findings earlier in the paragraph and then to have explained why the results differed.
655 But is it not the case that trees grown in the open tend to be shorter for a given DBH than those grow in closed conditions? Certainly that behavior is used by silviculturists to control the form of trees.
661 But why are study objectives what is being considered? Do the authors mean hypotheses? These are not the same.
663 but if one is protecting species diversity isn’t one protecting from loss? My point is that this seems to be repetitive.
759 Not sure what it means for a tree to outgrow its surroundings. Is that even possible?
Comments on the Quality of English Language
See specific comments
Author Response
While the revisions of the authors have improved the manuscript and in some cases provided much needed detail to understand the study, the manuscript is still problematic as written. The main points of concern:
1 It is still hard to find a logical flow in the introduction. Points appear over and over in different paragraphs.
Answer: Thanks for the comment. The introduction is modified to get a logical flow
2 The justification for the study also remains weak. It seems to be that because these relationships were not studied in this particular location that it needs to be studied. That may be, but that is largely a local and not a global justification.
Answer: Thanks for the comment. The justification is edited to clarify the global justification for doing this research.
3 The discussion is not compelling nor is it logically arranged. Again as with the introduction points seem to be repeated in different paragraphs. And the importance and significance of the findings is not made clear.
Answer: Thanks for the comment. We have revised the discussion section thoroughly and logically.
The methods described seem to only deal with standing dead wood. How was downed dead wood measured or was it excluded? If excluded the authors need to make sure their terminology regarding results reflects that.
Answer: Thank you for your comment, based on your comment we have added the subheading 2.2.3 in Materials and Methods section to address the question that how the deadwood samples were collected and calculated for biomass.
Specific comments (line)
63-65 The meaning of this sentence is not clear. What declines? The species? The peak performance? How can a peak performance decline as suggested? Also, how do species diversity and vegetation diversity differ? How are they different?
Thanks for the suggestion. The sentence is corrected and the clarity is improved.
112 richness is an index of biodiversity. Biodiversity can be described using richness, evenness, and heterogeneity. So this statement does not make sense as written.
Thanks for pointing this out. The sentence is restructured and written.
149-150 This is a study objective. It should therefore go with the other study objective. It would also be helpful to have a hypothesis related to this objective so that it matches the other study objective. Given the lack of background on this topic in the introduction it would also make sense to have at least one paragraph on this topic in the introduction. Otherwise it seems likes something that was just added on.
This is added with the other study objectives and a related hypothesis is given. A paragraph on this topic is also added to the introduction section.
279 It is not clear how this method could determine soil bulk density. It could certainly be used to determine the mass in a certain size fraction.
Thanks for the suggestion. The method for determining the soil bulk density is explained in section 2.4.
300 It is not clear what “deliberated” means in this context. Do the authors mean “determined”?
Thanks for correcting this. We have corrected this word in the manuscript
599-600 This sentence is not needed. It should be placed in the methods to justify the use of PCA.
The sentence is removed and added to the method section.
635 As noted before, if trees are present, then if must be that there is sufficient resources for tree growth. The authors must be referring to something else, but as written this is circular logic.
Thanks for pointing this out. The above mistake is corrected in the manuscript.
647 It would have been more interesting to contrast the different findings earlier in the paragraph and then to have explained why the results differed.
Thanks for the comment. The manuscript is edited based on the above comment.
655 But is it not the case that trees grown in the open tend to be shorter for a given DBH than those grow in closed conditions? Certainly that behavior is used by silviculturists to control the form of trees.
Thanks for the comment. We have added a few research that supports our finding
661 But why are study objectives what is being considered? Do the authors mean hypotheses? These are not the same.
Thanks for the suggestion. This is corrected in the manuscript.
663 but if one is protecting species diversity isn’t one protecting from loss? My point is that this seems to be repetitive.
Thanks for the comment. This is corrected in the manuscript.
759 Not sure what it means for a tree to outgrow its surroundings. Is that even possible?
Thanks for pointing this out. This is corrected in the manuscript.
Round 3
Reviewer 2 Report
Comments and Suggestions for Authors
Although the authors improved the manuscript, there are still some errors which need to be corrected. Since these errors are easily corrected, I suggest only minor revisions.
1st, there is still a misunderstanding throughout the paper what a gradient is. In the data there is only one altitudinal gradient consisting of 5 altitudinal classes. Thus, not the gradients have an influence on something, but the gradient (or the altitudinal classes) has (have) an effect on ….
2nd, the equation for the Simpson index is still wrong. It is D=1-∑pi². Since pi<1 by definition, 1/pi must be >>1. The results show, that the authors probably calculated the Simpson index correctly, only the equation is written wrong.
3rd, Throughout the paper the term “tree count” should be replaced by “number of trees per ha”.
In the abstract there is still an R² = -0.7
In line 368 the test the authors used war “Tukey’s post hoc test”.
In Table 2 The authors should add in the header or in the footnote: "B is the regression coefficient, beta is the standardized regression coefficient".
Comments on the Quality of English LanguageThere are only a few corrections which I suggest in my comments for the authors
Author Response
1st, there is still a misunderstanding throughout the paper what a gradient is. In the data there is only one altitudinal gradient consisting of 5 altitudinal classes. Thus, not the gradients have an influence on something, but the gradient (or the altitudinal classes) has (have) an effect on ….
Answer: Thank you for your comment, we have revised and replaced altitude with altitudinal gradients to address well to this query. Hope this address the question adequately, also conveys and present the clear message to the readers.
2nd, the equation for the Simpson index is still wrong. It is D=1-∑pi². Since pi<1 by definition, 1/pi must be >>1. The results show, that the authors probably calculated the Simpson index correctly, only the equation is written wrong.
Answer: Thanks for the comment, equation has been revised as suggested.
3rd, Throughout the paper the term “tree count” should be replaced by “number of trees per ha”.
Answer: Replaced.
In the abstract there is still an R² = -0.7
Answer: Corrected.
In line 368 the test the authors used war “Tukey’s post hoc test”.
Answer: Corrected.
In Table 2 The authors should add in the header or in the footnote: "B is the regression coefficient, beta is the standardized regression coefficient".
Answer: Added as a footnote.
Reviewer 3 Report
Comments and Suggestions for Authors
General comments
The manuscript needs to be carefully proofread as there a many, many issues regarding punctuation, word spacing, etc.
There are also cases in which the wrong word is used as noted below.
Specific comments (line)
36 The period needs to go after the last word in a sentence
21-46 there are numerous cases in which spaces are missing between words.
51 I still do not understand this sentence. What is the difference between species diversity and vegetation diversity?
55 Do the authors mean topographic instead of geologic? These are not geological factors, they are topographic factors.
573 I am not sure that the relationship between height and diameter proves this.
596 none of these variables are growth variables per se. Growth involves a time interval, otherwise it is meaningless.
627 Do the authors mean hypotheses instead of objectives? And objective is something one wishes to achieve or do. So how can a result be consistent with a study objective?
635 Again, I am not following the logic here. Do the authors mean hypothesis?
Author Response
The manuscript needs to be carefully proofread as there a many, many issues regarding punctuation, word spacing, etc. There are also cases in which the wrong word is used as noted below.
Answer: Thank you for your comments, we have revised the MS thoroughly.
Specific comments (line)
36 The period needs to go after the last word in a sentence
Answer: Corrected.
21-46 there are numerous cases in which spaces are missing between words.
Answer: Corrected.
51 I still do not understand this sentence. What is the difference between species diversity and vegetation diversity?
Answer: We have revised the sentence.
55 Do the authors mean topographic instead of geologic? These are not geological factors, they are topographic factors.
Answer: Thank you for your comment, we have revised it.
573 I am not sure that the relationship between height and diameter proves this.
Answer: We have removed the statement to avoid any misunderstanding.
627 Do the authors mean hypotheses instead of objectives? And objective is something one wishes to achieve or do. So how can a result be consistent with a study objective?
Answer: Thank you for your comments, corrected.
635 Again, I am not following the logic here. Do the authors mean hypothesis?
Answer: Thank you for your comments, corrected.